# Snail-inspired water-enhanced soft sliding suction for climbing robots

Tianqi Yue [1], Hermes Bloomfield-Gadêlha [1] & Jonathan Rossiter [1] ✉

Snails can stably slide across a surface with only a single high-payload sucker, offering an efficient adhesive locomotion mechanism for next-generation climbing robots. The critical factor for snails' sliding suction behaviour is mucus secretion, which reduces friction and enhances suction. Inspired by this, we proposed an artificial sliding suction mechanism. The sliding suction utilizes water as an artificial mucus, which is widely available and evaporates with no residue. The sliding suction allows a lightweight robot (96 g) to slide vertically and upside down, achieving high speeds (rotation of 53°/s and translation of 19 mm/s) and high payload (1 kg as tested and 5.03 kg in theory), and does not require energy during adhesion. Here, we show that the sliding suction is a low-cost, energy-efficient, high-payload and clean adhesive loco-motion strategy, which has high potential for use in climbing robots, outdoor inspection robots and robotic transportation.

Adhesion strategies, by which robots directly generate the attractive force between an object and a substrate, have been developed to allow for robots to grip, manipulate and climb[1]. Climbing robots have been developed with various adhesion strategies, including suction[1,2], electromagnetic adhesion[3,4], gecko adhesion[1,5] and electro-adhesion[1,6]. The typical target for these climbing robots is maximising the adhesive force and, hence, the holding or perching security of the robot. However, the higher the adhesive force generated, the higher the induced friction force between the adhesive pad and the substrate. To operate under the restriction of a high friction force, climbing robots must have multiple adhesive pads and intermittently detach, move and reattach the pads to climb, resulting in a discrete "walking" movement[3,7–10]. This discrete gait has several shortcomings: the control system must frequently break and regenerate the adhesive force on each pad; a complex transmission structure must be built into the robot, adding weight; the repositioning of the adhesive pad and the regeneration of the adhesive force waste energy; and the total adhesive force is significantly lower during movement since some of the adhesive pads are detached.

In nature, snails utilise a different climbing strategy—the sliding suction. Snails can maintain a constant high suction force and simultaneously slide on the substrate, without the need for multiple adhesive pads[11,12]. This contradicts the stereotype that a regular suction cup

is difficult to slide on the substrate and is susceptible to falling off when applied by a tangential force. However, a common phenomenon —a cup can easily slide on a wet table with strong suction—indicates that the suction and sliding movement of suction cups are not a conflict. The liquid film at the suction interface is the common feature found between snails (i.e., mucus) and the "sliding cup" (i.e., water). As reported by several studies, a thin liquid film could reduce the friction between two contacting surfaces[13] and enhance the suction[14]. Therefore, we believe that the critical challenge for designing a snail-like sliding suction cup involves materials and structure design for intelligently introducing the liquid film into the suction interface while maintaining a compliant and effective suction.

Several related works have been reported which aim to deliver stronger and more versatile suction adhesion. To eliminate the contact friction force, one proposed solution is to lift the suction pad up from the substrate[15,16]. However, due to the introduced gap between the suction pad and the substrate, leakage is severe, and a centrifugal pump must be used to constantly generate a vortex and maintain the suction. This consumes too much energy and is noisy. Shi and Li replaced air with water as the vortex medium, thereby achieving higher adhesive force, but the introduced sucker-substrate gap still remained a critical cause of suction leakage[17]. Chan et al. developed a gel-enhanced snail-like robot which can climb on a slope[18]. However, the

[1]School of Engineering Mathematics and Technology, and Bristol Robotics Laboratory, University of Bristol, Bristol, UK.
✉e-mail: Jonathan.Rossiter@bristol.ac.uk

adhesion was solely achieved—and limited—by the stickiness of the gel, and upside-down adhesion was not reported. Our former snail-inspired robot achieved snail-like sliding but lacked adhesion functionality[19]. Miyake et al. demonstrated that silicon oil can lubricate the suction interface and improve suction performance on rough surfaces[20,21]. However, this concept did not achieve autonomous long-distance sliding suction by a robot in a dry environment and the persistence of silicone oil can contaminate the substrate.

In this work, a water-enhanced sliding suction mechanism is proposed. In the next section, we introduce the method and principles for achieving water-lubricated sliding suction, demonstrate sliding suction performance and outline the fabrication method. We then introduce the design of the sliding suction robot, its physical model and the sliding suction experimental results including rotation tests, translation tests, loaded sliding tests, wall-climbing tests and user-controlled upside-down sliding tests. We then discuss experimental results and issues encountered and review the unique features of the sliding suction robot and propose future improvements. Finally, we conclude the work, highlight its significance and contribution, and present potential applications.

## Results

### Principle of sliding suction mechanism

Since suction is generated by a pair of contacting surfaces (i.e., the suction cup and the substrate), friction at the suction interface is the dominating factor in determining if a suction cup can slide. Snails secrete mucus to lubricate the contacting interface and reduce the

coefficient of friction (CoF), making sliding movement possible. In addition, mucus can enhance the suction by sealing the gaps between the abdomen and the substrate[22]. Coordinated body movements, in the form of travelling waves, then generate the driving force to slide forward. Other low-viscous liquids, such as water, also reduce friction, which previous studies in contact and friction mechanics have well reported[13]. Replicating the role of snail mucus, the liquid stops the two surfaces making direct contact, acting as a mechanical lubricant, and smooths the microscopic local surface roughness[13]. In the meantime, the liquid also enhances both suction strength and longevity in comparison with a dry environment[14]. We conclude from the above that both friction reduction and suction enhancement can be achieved simultaneously by applying liquid to the suction interface. By mimicking the soft single-sucker structure and the liquid exudation mechanism of the snail, effective sliding suction can be achieved in artificial suction cups and, by extension, suction-based climbing robots.

### Designing water-enhanced sliding suction mechanism

To achieve artificial sliding suction like a snail, the contact interface should be filled with liquid, with no dry-contact areas. Snails spread the mucus to the whole abdomen by cilia beating[23] (Fig. 1A). Cilia-driven spreading cannot be readily achieved in an artificial suction cup, and therefore a water self-spreading method is employed. We use hydrophilic silicone, which is fabricated by adding poly(dimethylsiloxane-b-ethylene oxide) (PBP, a hydrophilic polymer) into the plain silicone[24], as the material of the suction cup's bottom pad which generates a strong attractive force to water molecules (Supplementary

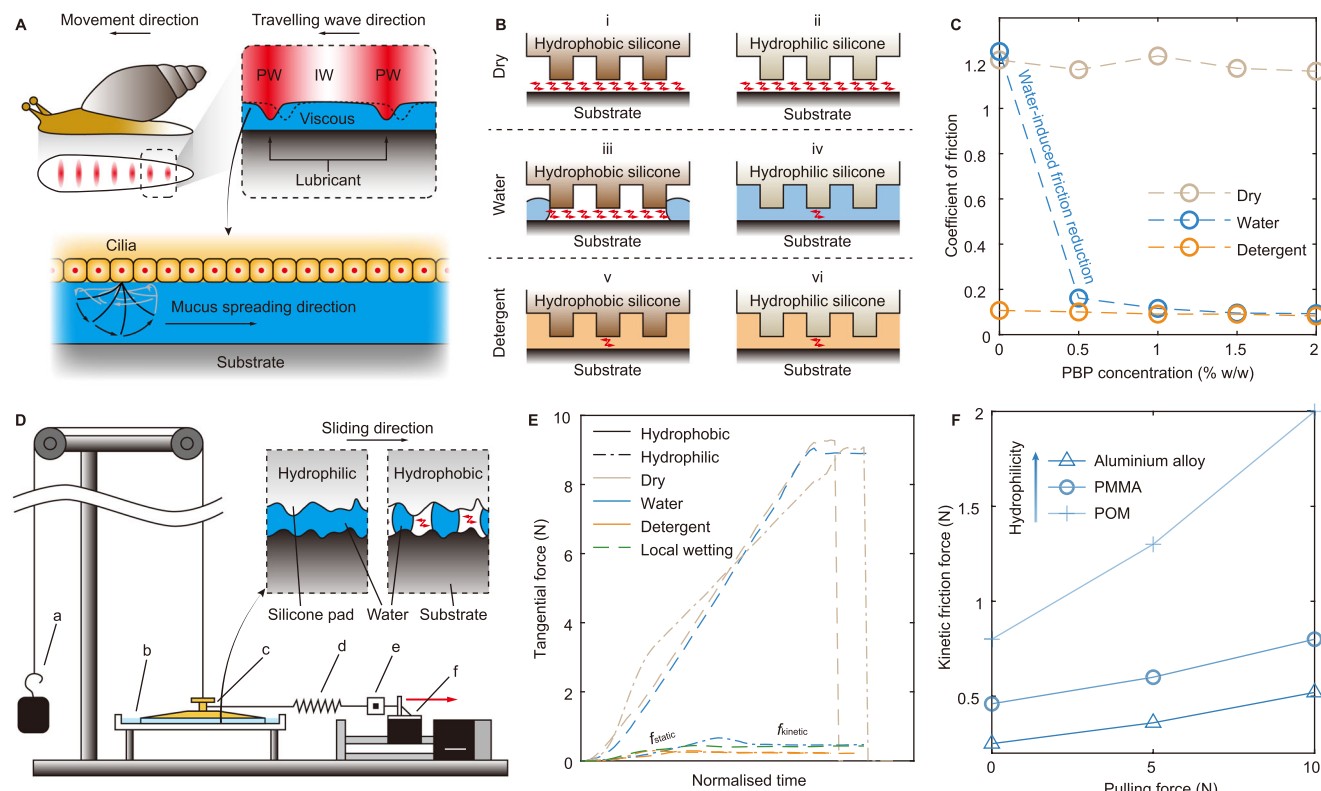

**Fig. 1 | The sliding suction mechanism. A** The snail achieves sliding suction by mucus secretion and muscular contraction. IW: interwave. PW: pedal wave. **B** Diagrams of dry and lubricated contact interfaces for unmodified (hydrophobic) and modified (hydrophilic) silicone. **C** Measured CoF with increasing PBP concentration in three friction conditions. **D** The experimental setup for evaluating sliding suction. a: mass. b: water tank with PMMA substrate. When wet conditions are required, the water tank is filled with shallow liquid - 3 mm, which just immerses the suction interface. c: the tested suction cup. d: tensile spring. e: load cell, for

measuring the tangential force. f: linear stage, for applying a tangential force on the suction cup top. Insert: difference of wetting between the hydrophilic and hydrophobic silicone pads submerged in water. **E** Tangential force measured by the load cell during the sliding. The seven tests have different durations due to different friction values, and therefore we adjusted their time period to the same length for clarity (labelled as normalised time). **F** Measured kinetic friction force as a function of the perpendicular pulling force of the $S_{HydPhi}$-water case on different substrates. Kinetic friction force corresponds to the flat period denoted by $f_{kinetic}$ in (**E**).

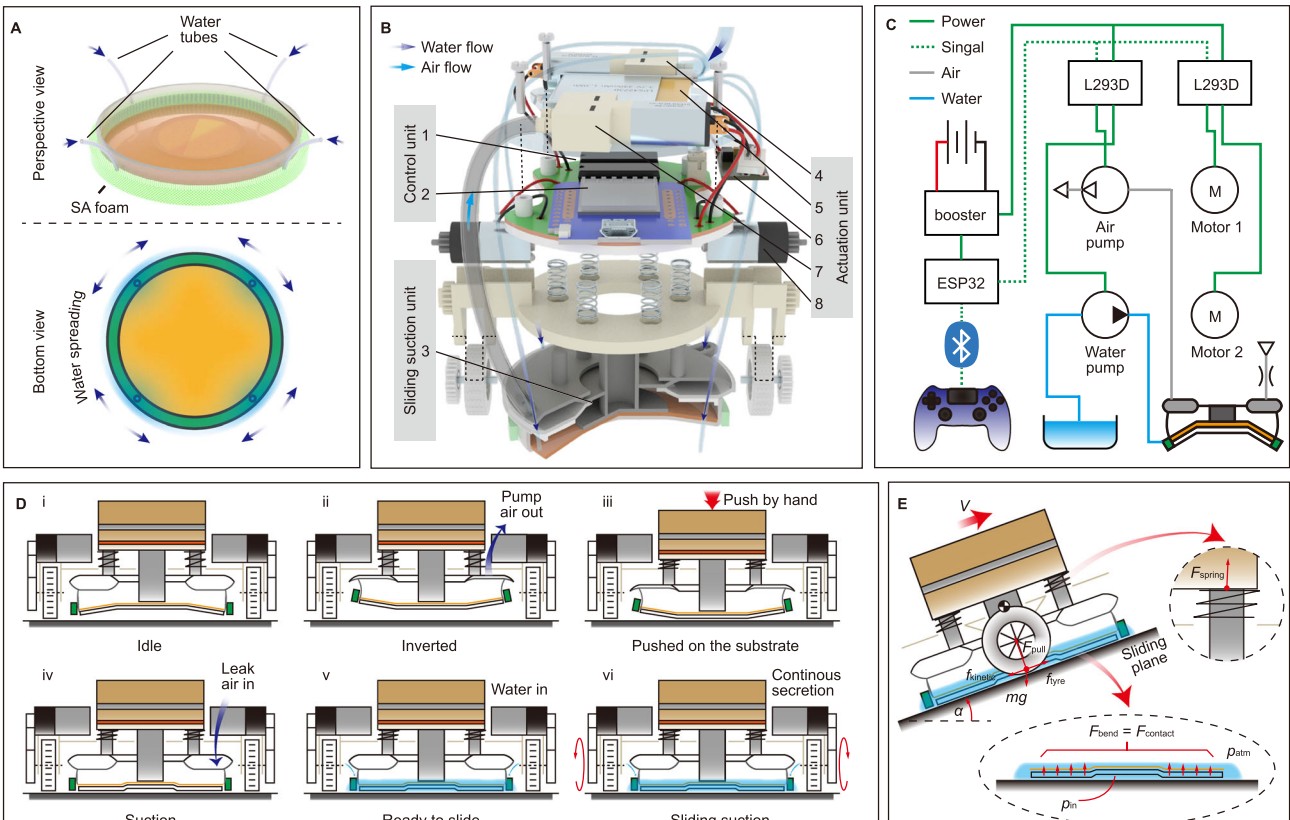

**Fig. 2 | Design of the SSR. A** The design of water secreting system. **B** The main structure of SSR. 1: motor/pump driver. 2: ESP32 development board. 3: sliding-suction cup. 4: water pump. 5: battery. 6: voltage booster board. 7: air pump. 8: motor. **C** Diagram of electric, pneumatic and hydraulic circuit of the SSR. **D** Working flow of the SSR. **E** Diagrams of the SSR physical model. Right insert: compressive force on each spring. Bottom insert: force applied at the suction interface.

Information S1). Once the hydrophilic silicone contacts with the substrate (the gap is small enough), water spontaneously spreads into the contact interface by capillary action—mimicking the cilia-enabled mucus spreading in the snail—and thus friction is reduced (Fig. 1B iv). In contrast, the plain silicone pad remains naturally hydrophobic and repels water from the interface, even if the hydrophobic silicone is placed on a pre-wetted surface. This is because the instantaneous submergence in water cannot let the tiny air bubbles trapped in the silicone surface disappear; therefore, these bubbles still remain between the silicone and the substrate, making the interface more like a dry contact and generating a high friction force (Fig. 1B i to iii). Alternatively, we could also leave the silicone pad hydrophilic and instead use a detergent solution, which exhibits strong attractive force to a wide range of surface materials, as the lubricant. However, the detergent solution lubricates both hydrophilic and hydrophobic silicone (Fig. 1B v and vi), making it difficult to utilise the friction difference between the hydrophilic silicone (used as the suction cup bottom) and hydrophobic silicone (used later as drive wheels in the robot). In addition, the detergent solution is more expensive and less readily available than water and will leave a residue upon drying.

### Demonstration of water-induced friction reduction

Flat samples of hydrophobic silicone (0% PBP) and hydrophilic silicone (0.5–2% PBP) were fabricated and their CoF on PMMA was measured under three conditions: dry, wetted by water and wetted by detergent solution (5% w/w). As shown in Fig. 1C, in dry condition, the increase of PBP concentration does not influence CoF ($\mu_{dry\_pho} \approx \mu_{dry\_phi} \approx 1.2$). Detergent solution significantly lubricates both hydrophobic silicone and hydrophilic silicone ($\mu_{det\_pho} \approx \mu_{det\_phi} \approx 0.1$). When water is used, a significant friction reduction up to 94%, can be seen between the

hydrophobic silicone ($\mu_{wat\_pho} \approx 1.23$) and hydrophilic silicone ($\mu_{wat\_phi} \approx 0.1$). As the concentration of PBP increases, the CoF approaches that of the detergent solution (for 2% PBP, $\mu_{wat\_phi} \approx 0.1$), indicating the interface is well lubricated. Increasing the PBP concentration above 2% does not significantly reduce CoF further, therefore 2% PBP is determined to be the optimum concentration. This test demonstrates the effectiveness of the materials-enhanced water self-spreading mechanism for reducing CoF. Subsequently, the artificial sliding suction cup will be designed based on this fundamental mechanism.

### Demonstration of water-enhanced sliding suction

Based on the proposed water self-spreading mechanism, a sliding suction cup as shown in Supplementary Fig. 2 can be designed. The sliding suction cup has a top layer (i.e., the polyurethane (PU) reinforcement layer) and a bottom layer (i.e., the PBP-silicone pad). According to our tests presented in Supplementary Information S2, slope angle of 10° and diameter of 50 mm are the optimal geometry to generate sliding suction. Two suction cups, one with hydrophobic silicone pad ($S_{HydPho}$) and another with hydrophilic (2% PBP) silicone pad ($S_{HydPhi}$), were fabricated. The detailed fabrication methods of the reinforcement disc and silicone pads are given in Supplementary Information S3. To test the sliding suction, the setup in Fig. 1D was used. $S_{HydPho}$ and $S_{HydPhi}$ were tested in three conditions: dry, immersed in water and immersed in detergent solution (5% w/w), and all with zero payload. In all three conditions, $S_{HydPho}$ and $S_{HydPhi}$ were successfully pulled into slide motion by the linear stage, at the same time maintaining suction, as can be seen in Supplementary Movie 1. The real-time tangential force recorded by the load cell is shown in Fig. 1E. In the dry condition, both $S_{HydPho}$ and $S_{HydPhi}$ slid only a little distance, then broke away from the substrate. When they were

immersed in liquid, none of them showed suction failure, demonstrating the compatibility of suction and sliding movement, and the suction safety improved by the liquid environment. $S_{HydPhi}$-water case results in a friction force reduction up to 95% (close to detergent-lubricated cases) compared to the $S_{HydPho}$-water case. The tangential force of all six combinations shows an initial increase, indicating that the suction cup is initially stationary and affected by the static friction force $f_{static}$. At some point the tangential force exceeds the maximum static friction force and the suction cup starts to slide and affected by a constant kinetic friction force $f_{kinetic}$. We denote $f_{static}$ and $f_{kinetic}$ in Fig. 1E, take the $S_{HydPhi}$-water case (bold line) as an example. $f_{kinetic}$ is the friction force that needs to be overcome during the sliding.

### Influence of perpendicular pulling payload and substrate materials on sliding friction

We further used 0, 500 and 1000 g mass to generate 0, 5 and 10 N perpendicular pulling force on $S_{HydPhi}$ in water environment, to see how the perpendicular pulling force affects the sliding friction. We performed the experiments on substrates made from three different materials with increasing hydrophilicity: polyoxymethylene (POM, water contact angle (WCA) = 76.8°), PMMA (WCA = 68°) and aluminium alloy (WCA = 57°). Results are shown in Fig. 1F. Increasing the pulling force from 0 to 10 N only slightly increases $f_{kinetic}$ on all three substrates. In particular, the $f_{kinetic}$ on POM is the largest and on aluminium alloy is the smallest. This relation is inverse to the hydrophilicity of the three materials, which will be discussed later. In the rest of the paper, we will use PMMA as the default substrate material if not otherwise specified. $f_{kinetic}$ increases from 0.46 N to 0.8 N on PMMA substrate, indicating that the water-enhanced $S_{HydPhi}$ can easily slide when it is strongly pulled, in other word, heavily loaded. This can be explained by the force analysis[25,26]:

$$f_{kinetic} = \hat{\mu}_{wat\_suc} F_{contact},$$
$$F_{contact} = F_{bend}, \qquad (1)$$
$$F_{bend} = (p_{atm} - p_{in})A - F_{pull} + mg,$$

where $m$ is the mass of the suction cup. Here we assume the suction cup is place on a horizontal substrate, and will discuss the situation when the substrate is tilted later. $\hat{\mu}_{wat\_suc}$ is the normalised CoF (will be explained later) at the suction interface between the wet $S_{HydPhi}$ and PMMA sheet, $p_{atm}$ and $p_{in}$ are the atmospheric pressure and the pressure inside of the suction region respectively, $A$ is the area of the sealed suction region, $F_{contact}$ is the contact force between the suction cup and substrate, which is equal to its counterforce $F_{bend}$, the force caused by the suction cup bending (i.e., deformation), and $F_{pull}$ is the perpendicular pulling force. This increase of $F_{pull}$ slightly causes the increase of $F_{bend}$; therefore, it causes a tiny increase on $f_{kinetic}$ after $F_{contact} = F_{bend}$ multiplying by $\hat{\mu}_{wat\_suc}$. We will explain the details of this phenomenon later. This is a crucial property of sliding suction: it indicates that a sliding suction cup can achieve heavy-loaded sliding by only increasing sliding force (and hence energy input) a little—since the adhesive suction force passively increases by the pressure differential which consumes no energy, while the $f_{kinetic}$ to be overcome just increases slightly. This is an important advantage of sliding suction and contrasts markedly with other adhesion strategies which must consume more energy when loaded, e.g., EMA and EA. This reveals the potential for high load capacity of our sliding suction cups.

### Creating a local water environment on dry land

For practical applications, we must consider when the suction cup is applied on dry substrates; therefore, a local water environment must be created complying with the sliding suction. We design a water secreting system as shown in Fig. 2A. Given the difficulty of mimicking snails' mucus spreading via cilia beating (Fig. 1A), we use a super-absorbing (SA) foam to enclose the suction cup's rim. Four thin

silicone tubes are located around the rim to transport water from a water pump (4 in Fig. 2B) to the SA foam, then the high capillary force of the SA foam enables water to be evenly spread around the suction cup rim; finally, water can be spontaneously adsorbed into the suction interface by the capillary force induced by the hydrophilic silicone pad. This water secreting system allows the robot to benefit from a local water environment (with water supply rate of approximately 0.05 mL/s when the SSR slides 19 mm/s, according to the later physical model and experimental results), achieving a same friction reduction as submersion in a water tank. This is shown as the 'local wetting' condition in Fig. 1E.

### Designing a sliding suction robot

We design a sliding suction robot (SSR) to demonstrate the practicality of the sliding suction. Since the details of robot design is not the focus of this work, we propose its main structures in Fig. 2B, and provide design and structural details in Supplementary Information S2. The SSR mainly consists of a sliding suction unit, an actuation unit and a control unit. The sliding suction unit contains a same sliding suction cup with SA foam as we described before and a snapping chamber (will be explained later). The actuation unit includes pumps, tubes, motors and wheels to supply water, activate suction and drive the movement. A portable water tank (not shown in Fig. 2B) can be mounted on the back of the robot (shown in Supplementary Fig. 8A); therefore, the SSR can be completely untethered. The control unit is an ESP32 board which can be remotely controlled by a game controller via Bluetooth. The diagram of the SSR system is shown in Fig. 2C.

### Working principle of the sliding suction robot

The SSR achieves sliding suction through following steps (Fig. 2D). Before attaching to the substrate, the user manually wets the suction cup surface and the SA foam with water. Initially, the SSR is in a relaxation state (Fig. 2D i). When the SSR is about to attach to the substrate, the air pump (7 in Fig. 2B) works for 2 seconds to actuate the snapping chamber and transitions the SSR to an inversion state by lifting the suction cup rim (Fig. 2D ii). Detailed explanation is provided in the Supplementary Information S2. Then the air pump stops working, and the user manually holds the SSR against the substrate and maintains a contacting force -5 N (Fig. 2D iii). Air slowly leaks into the snapping chamber through a tiny through-hole (diameter of 0.5 mm) on the lid, until the snapping membrane snaps forward to transition the SSR to a suction state (Fig. 2D iv). This period takes -3 seconds then the user can release and now the SSR has safely attached to the substrate. Six radially placed springs provide the compressive force for the rubber wheels to generate static friction. Next, the water pump supplies water for 3 seconds to form the local water seal across the contact interface (Fig. 2D v). After this step, by controlling the rotation direction and speed of two wheels, the SSR can slide on the substrate (Fig. 2D vi). Water is continuously supplied during the sliding to compensate for the loss on the substrate. To detach, the air pump works for 2 s to lift the suction cup rim, thereby releasing suction. The water film left on the substrate finally evaporates leaving no residue. A complete working flow can be seen in Supplementary Movie 2.

### Physical models of the SSR

Here, we provide critical formulae in this section and put detailed derivations in Supplementary Information S5. Key parameters are the sum of the static friction force on two tyres, $f_{tyre}$, and the kinetic friction force on the suction interface, $f_{kinetic}$. $f_{tyre}$ and $f_{kinetic}$ are in the same plane (defined as the "sliding plane", as shown in Fig. 2E). We assume the SSR is sliding with constant velocity $V$ on a flat substrate (tilt angle $\alpha$). The influence of tilting moment (i.e., the mass of the robot times the distance of centre-of-mass from the substrate) is neglectable. This is because it affects the contacting pressure distribution but not the total contacting force at the suction interface and

we assume the CoF is constant and not affected by the pressure change. In addition, the robot is lightweight (96 g), has low centre-of-mass (~20 mm) and a relatively large suction cup (diameter of 50 mm), further reducing the influence of the tilting moment. $f_{tyre}$ can be derived by the tangential force balance,

$$\vec{f}_{tyre} + \vec{f}_{kinetic} + m\vec{g} \cdot \sin\alpha = 0, \tag{2}$$

where $m = 0.096$ kg is the mass of the SSR. Now $f_{kinetic}$ is the only unknown parameter in the model, which can be calculated via $f_{kinetic} = \hat{\mu}_{wat\_suc} F_{bend}$ in Eq. (1). Through Fig. 1F we find that $f_{kinetic}$ is influenced by $F_{pull}$; therefore, we know $F_{bend}$ is affected by $F_{pull}$ since $\hat{\mu}_{wat\_suc}$ is constant. $F_{bend}$, the bending-induced force when a suction cup is adhering to the substrate, can be measured by steps described in Supplementary Information S6. We measured $F_{bend} = 1.6$ N when $F_{pull} = 0$ N. According to the result in Fig. 1F,

$$\hat{\mu}_{wat\_suc} = \frac{f_{kinetic}(F_{pull} = 0)}{F_{bend}(F_{pull} = 0)} = 0.29 = const. \tag{3}$$

We observe that the calculated normalised CoF of the sliding suction cup $\hat{\mu}_{wat\_suc} = 0.29$ is greater than the measured CoF of the flat silicone pad $\mu_{wat\_phi} = 0.1$ (Fig. 1C), although they employ the same contacting materials—hydrophilic silicone and PMMA—in the same water-contact condition. The relation between $\hat{\mu}_{wat\_suc}$ and $\mu_{wat\_phi}$ is

$$\hat{\mu}_{wat\_suc} = \frac{f_{kinetic}}{F_{contact}} = \frac{\iint \mu_{wat\_phi}(s)p(s)\mathrm{d}s}{F_{bend}}, \tag{4}$$

where $p$ is the contacting pressure and $S$ is the contacting region between the suction disc and the substrate. Several reasons cause the difference: First, the suction cup's silicone pad was cured in closed moulds, while the flat silicone pad was cured in open moulds; therefore, the suction cup's bottom surface is rougher than the flat silicone pad. Second, the non-flat geometry of the suction cup influences the surface micro-topography and therefore air is not completely squeezed out of the suction interface during the adhesion, which generates unwetted sites and adds static friction force. The measured normalised CoF on hydrophobic wheel tyres in the water-contact case shows the same trend: $\hat{\mu}_{wat\_tyr} = 5.56 > \mu_{wat\_pho} = 1.23$. However, just as $\mu_{wat\_phi} \ll \mu_{wat\_pho}$, $\hat{\mu}_{wat\_suc} \ll \hat{\mu}_{wat\_tyr}$ enables sufficient wheel grip to initiate and maintain sliding suction.

According to $f_{kinetic} = \hat{\mu}_{wat\_suc} F_{bend}$ in Eq. (1), once we got $\hat{\mu}_{wat\_suc}$ and the $f_{kinetic} - F_{pull}$ relation (Fig. 1F), the $F_{bend} - F_{pull}$ relation can be calculated as

$$F_{bend} = 0.12 F_{pull} + 1.54. \tag{5}$$

Equation (5) indicates that the increase of $F_{pull}$ slightly increases $F_{bend}$; therefore, $f_{kinetic}$ is further slightly influenced by $F_{pull}$. The decrease of the internal pressure $p_{in}$, which does not consume energy, balances most of the increase of $F_{pull}$, according to $F_{pull} = (p_{atm} - p_{in})A - F_{bend} + mg$ in Eq. (1). Although the decrease of $p_{in}$ makes the suction cup squeeze harder, it does not change the geometry very much; this is why $F_{bend}$ only increases slightly and the sliding suction mechanism has such high loading ability. Equations (1), (2), (3) and (5) are the critical formulae of the physical model of the SSR, through which we calculated the sliding ability, climbing ability and loading ability of the SSR. Details of the formula derivation are provided in Supplementary Information S5. Calculation results suggest that the SSR can easily slide upside-down, climb on the wall and carry a heavy mass.

In addition to the force analysis, another physical interaction occurring is the water secretion and loss during sliding. Optimally, the water secretion rate should always be equal to the water loss rate,

which is caused by the robot travelling and leaving a water film on the substrate. The sliding suction motion can be divided into rotation and translation. Ideal rotation results in no relative translation and thus no water loss, while translation leaves a thin water film on the substrate. Therefore, rotation should not need additional water secretion while translation does. The measured water film thickness was $t_{film} \approx 0.05$ mm on the PMMA sheet. Therefore, the water secretion rate in rotation ($Q_{rot}$) and translation ($Q_{trans}$) can be written as

$$\begin{aligned} Q_{rot} &= 0, \\ Q_{trans} &= V d t_{film}, \end{aligned} \tag{6}$$

where $d = 54$ mm is the diameter (including the SA foam rim) of the suction cup.

## Experimental setup

A PMMA sheet was hinged on the ground with adjustable tilted angle $\alpha$ to the horizontal plane. 0° (horizontal), 45°, 90° (vertical), 135° and 180° (upside down) were set as the substrate tilt angle for SSR to adhere. A camera recorded the movement of the SSR. When testing the suction cup on real-world surfaces, the user controlled the robot in real-time using the wireless game controller as we explain in Supplementary Information S8. In all tests, we applied 5 V to the two DC motors, therefore the SSR translated/rotated with an approximately constant velocity. The water secretion rate was based on calculations from the following physical model.

## Rotation test

The SSR was actuated to rotate clockwise for 5 seconds and repeat for 3 times on the tilted PMMA sheet. The experimental results are shown in Fig. 3A-C. Trajectories in Fig. 3B show that the SSR successfully achieves rotation movement. However, some deviation is observed which is caused by tyre slip and transient stiction (will be discussed later). To evaluate movement, two parameters (as denoted in Fig. 3A) are defined. Centre deviation (the distance from the final point to the starting point) and average rotational velocity are denoted as $d_{cen}$ and $\omega_{rot}$ respectively. $d_{cen}$ and $\omega_{rot}$ values of the five cases are shown in Fig. 3C. $d_{cen}$ is not large (smaller than the robot size) after all 5-second rotations, and the average rotational velocity $\bar{\omega}_{rot}$ of all tests is approximately 53°/s.

## Translation test

The SSR was actuated to translate for 2 s and repeat for 3 times on the tilted PMMA sheet. The robot was actuated to climb upward and downward respectively with each tilt angle. For 0° and 180° cases, we actuated the robot to slide horizontally and draw the horizontal trajectories on the ordinate axis in Fig. 3E for ease of comparison with other cases. Results of translation tests are shown in Fig. 3D-F. Movement deviation is also observed in Fig. 3E. To evaluate, we denote the average translational velocity and the offset distance after the 2-second translation as $V_{trans}$ and $d_{off}$ respectively, shown in Fig. 3D. $V_{trans}$ and $d_{off}$ of eight cases are shown in Fig. 3F. The average $\bar{V}_{trans}$ of all tests is 19 mm/s. Tilt angle of the substrate has an apparent influence on its translational sliding ability. We will discuss this later.

## Payload sliding test

A mass of 1 kg was hung under the robot, which generated approximately 11 N of pulling force, including the SSR mass. The SSR was then attached upside down to the PMMA sheet. The SSR was then actuated to rotate for 5 seconds and translate for 2 seconds. Time-lapse photos shown in Fig. 3G and Fig. 3H demonstrate that the SSR successfully achieves 5-second rotation and 2-second translation sliding with 1 kg mass loading. The measured $\omega_{rot}$ and $V_{trans}$ are 23.4°/s and 17.1 mm/s, respectively, which is slightly smaller than the non-load case. Videos of rotation, translation and loaded sliding tests can be seen in Supplementary Movie 3.

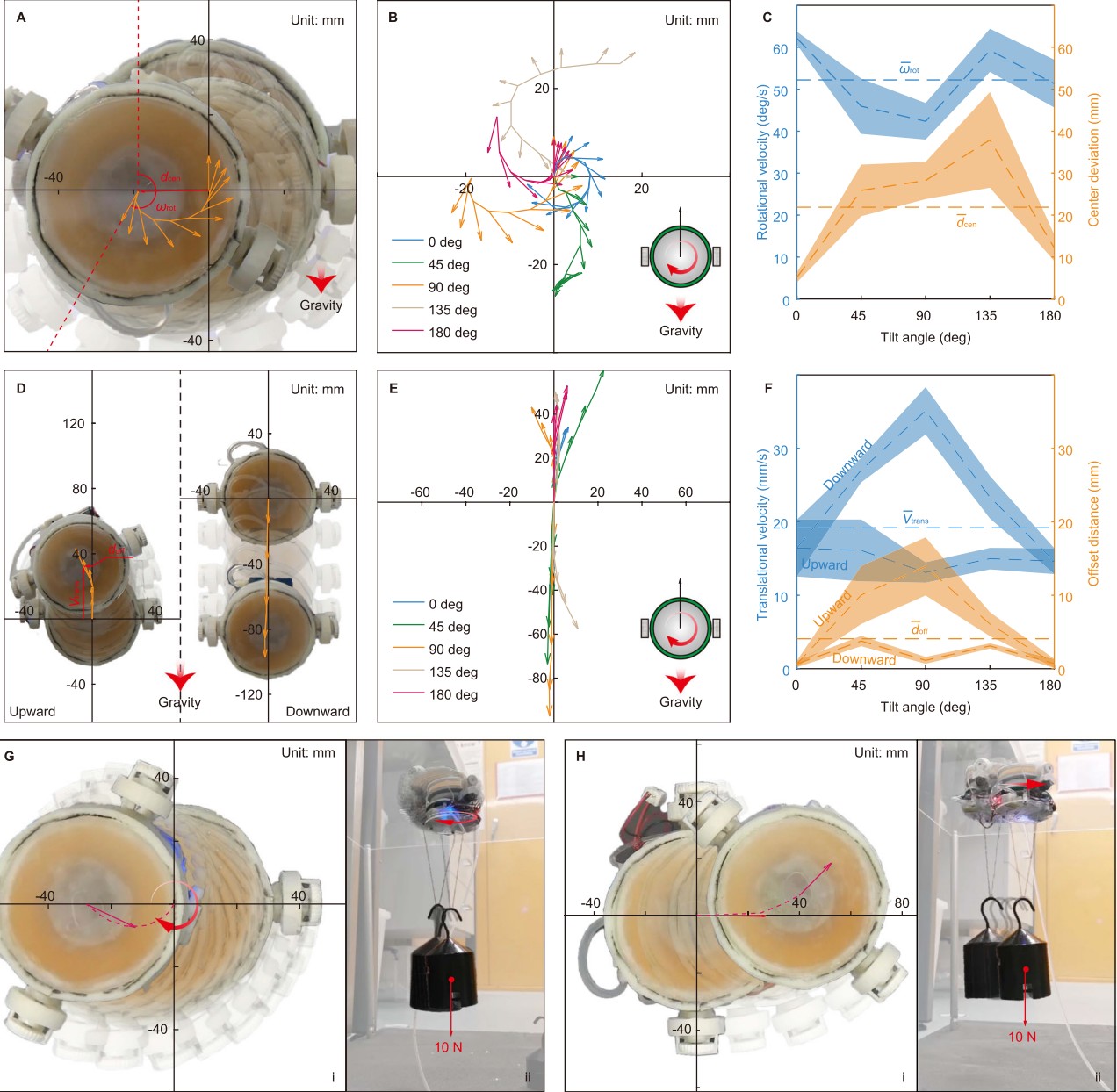

**Fig. 3 | Experimental characterisation of the sliding suction ability of the SSR.**
**A** Time-lapse photos of the rotation test on the vertically (90°) placed PMMA sheet.
**B** Rotational trajectories of the SSR centre on substrate with five different tilt
angles. Arrows indicate the head orientation. Position and orientation of the robot
are recorded at every 0.5 s. **C** $d_{cen}$ and $\omega_{rot}$ of the rotation tests. Dashed lines are the
average of corresponding parameters. Shadows are the standard deviation. **D** Time-
lapse photos of the translation test on the vertically placed PMMA sheet.
**E** Translational trajectories of the robot on substrate with five different tilted
angles. For the horizontal trajectories of 0° and 180° cases, we align movement with
the ordinate axis. **F** $V_{trans}$ and $d_{off}$ of the translation tests. **G** Time-lapse photos of
the upside-down (180°) loaded rotation test. **H** Time-lapse photos of the upside-
down loaded translation test.

## Water film evaporation

The SSR was actuated to slide upside-down a PMMA sheet as shown in
Supplementary Fig. 8A, leaving a thin water film on the substrate. The
water film faded quickly in the beginning 5 minutes, and totally dis-
appeared within 37 minutes leaving no residue on the substrate, as
shown in Supplementary Fig. 8B.

## Demonstration: untethered climbing

To demonstrate the practical application of the SSR, it was tested to
climb on two real-world wall surfaces, a glass wall and a painted metal
wall. The SSR was equipped with a portable water tank and subsequent
untethered. Figure 4A shows that the robot can climb upward both on
the glass window and the painted metal wall. During the experiments,

an offset from the desired vertical direction was observed, but was
real-time corrected by user control, thus the SSR can follow an upward
direction. We also observed that climbing on the glass window was
smoother than on the painted steel wall, as seen in Supplementary
Movie 4.

## Demonstration: user-controlled weight transportation with obstacle avoidance

We recruited three volunteers without knowledge of sliding suction.
Volunteers were taught the functions of the buttons on the game
controller, and were required to use the game controller to drive the
SSR (loaded with 200 g mass) and avoid obstacles from the starting
point to the destination. The diagram of experimental setup is shown

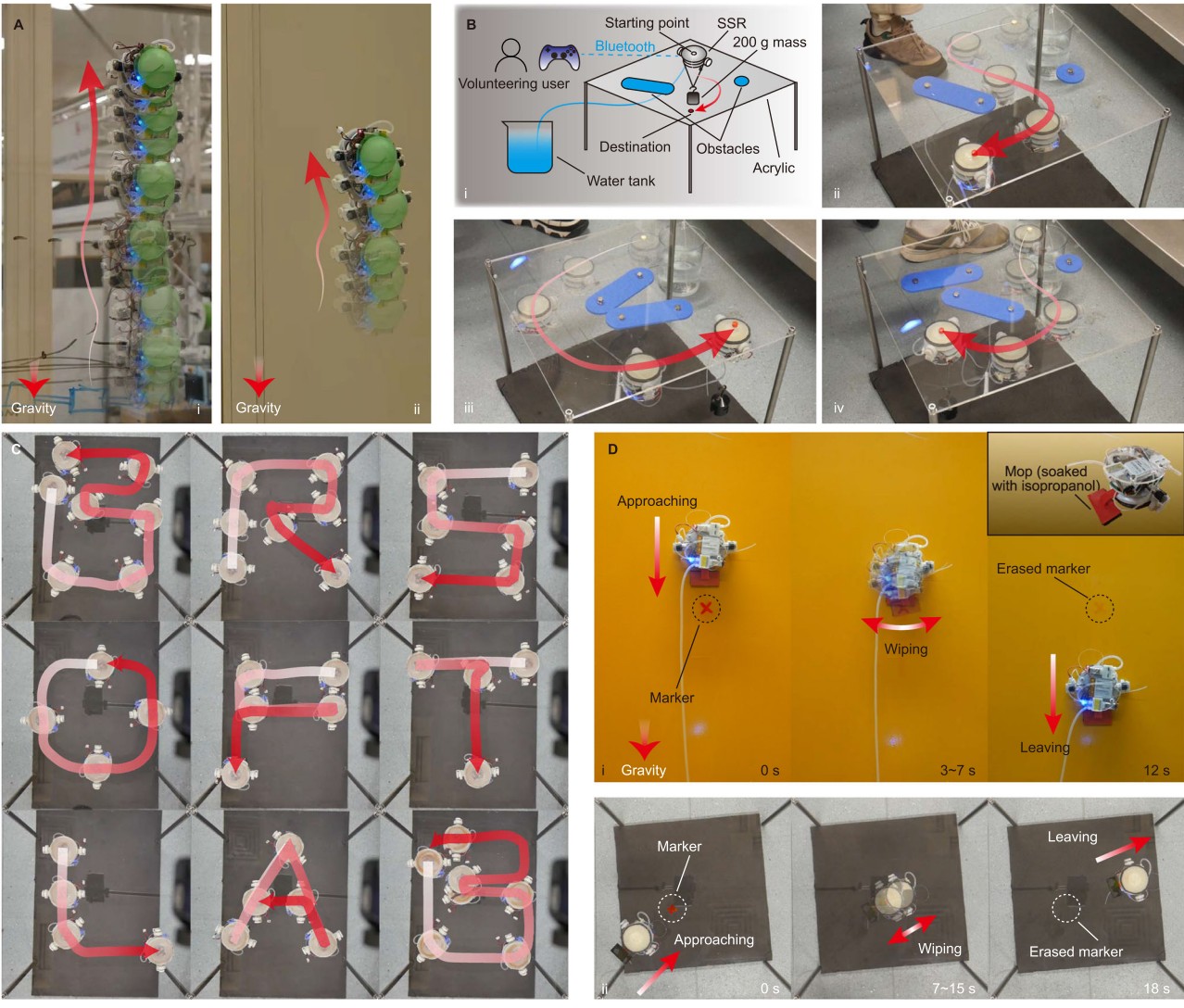

**Fig. 4 | Demonstration of applications of SSR. A** The upward climbing ability of SSR on real-world surfaces. i: on glass window. ii: on painted metal wall. **B** User-controlled upside-down sliding with 200 g payload and obstacle avoidance. i: diagram of the experimental setup. ii to iv: three volunteers successfully controlled the SSR to slide from the starting point to the destination. **C** User-controlled route tracking of nine letters. **D** Demonstration of using the SSR as a carrier to clean markers on an upright painted metal wall (i) and an upside-down PMMA ceiling (ii).

in Fig. 4B i. All volunteers successfully and smoothly completed the task, as shown in Fig. 4B ii-iv and Supplementary Movie 4.

### Demonstration: user-controlled route tracking

The SSR was manually attached upside-down on the PMMA sheet, then remotely controlled to follow a series of user-defined routes—9 English characters of "BRSOFTLAB" (Fig. 4C). During the experiments, tyre slip and stuck movement were observed, causing slight deviation from the desired route. These deviations were corrected in real-time by the user. Video recordings can be seen in Supplementary Movie 4.

### Demonstration: erasing marker on different surfaces

A mop (fabricated by attaching a sponge to a 3D-printed bracket, as shown in the insert of Fig. 4D i) was attached to the SSR. The mop was soaked with isopropanol. Two surfaces, an upright painted metal wall and a PMMA ceiling, were drawn with a cross using a permanent marker pen. The SSR was remotely controlled to carry the mop to approach the permanent marker and then erase it using the mop. The SSR successfully completed the task on both the upright wall and ceiling, as shown in Fig. 4D and Supplementary Movie 4.

## Discussion

### Stuck movement and tyre slip

Theoretically, when the two DC motors on the SSR are supplied with the same voltage (5 V in former experiments), the SSR should rotate around the centre ($d_{cen} = 0$) or translate straight ($d_{off} = 0$). However, movement deviations were observed in Fig. 3. The deviation may be caused by the manufacture error and mismatches of the motors, which are not closed-loop controlled. It may also be caused by local regions which are not sufficiently wetted at the suction interface, which results in dry contact, re-emergence of static friction and a temporary pause in movement until the transition to sliding kinetic friction is re-established. This occurs more when the SSR slides too fast, climbs upwards (Fig. 4A), slides on highly hydrophobic surfaces or high-CoF surfaces, or lacks water secretion. The deviation may also be caused by the tyre slip. We noticed that when the wheel contacts with water too much, the wheel's friction reduces. This might be because water slowly fills the rubber wheel surface asperities, and the air is squeezed out, making the hydrophobicity of wheels lose efficacy. It is important to note that the SSR still stays stably on the substrate even when tyre slip and stuck movement occur. This is an additional safety feature of the SSR.

## Movement deviation caused by gravity and weight of the robot

For rotation (Fig. 3C), the SSR achieves smaller centre deviation in balanced-gravity cases (0° and 180°) than unbalanced-gravity cases (45°, 90° and 135°). It was observed that some water flowed downward by the effect of gravity and wetted one or more of the tyres contacting the substrate. Imbalance of friction due to this wetting, and subsequent tyre slip, leads to movement deviation. The weight of the SSR ( ~ 90 g when the portable water tank is not equipped) does not show significant influence on the rotation movement deviation since no clear difference between 45°, 90° and 135° cases was observed. For translation (Fig. 3F), the SSR's speed and deviation are, however, affected by gravity and its weight. According to the calculation in Supplementary Information S5, it is clear that the weight of SSR adds extra resisting torque for the motors to overcome when the SSR climbs upward (sliding speed 0°≈180°>45°≈135°>90°), but reduces the motors' load when the SSR slides downward (sliding speed 90°>45°≈135°>0°≈180°). Gravity component acting against the sliding direction can preclude water from fully wetting the front edge of the moving robot, thereby increasing the likelihood of stuck movement as described above. In contrast, the gravity component acting in the sliding direction accelerates water to wet the head of the suction cup. This explains why the upward climbing has larger deviation than downward sliding. As a summary, due to the effect of gravity and weight, tilted surfaces add extra difficulties for the SSR to manoeuvre on, while upside-down ceilings and upturned floors are easier.

## Discussions on high payload ability

Equation (1) indicates that pulling the suction cup only slightly increases the kinetic friction force that the SSR must overcome. This feature allows SSR to lift and move a heavy mass (as long as it does not exceed the maximum suction force) without substantially increasing motor power. This explains how the SSR can easily carry a 1 kg mass while sliding upside down, which is 10 times heavier than the robot. We calculated in Supplementary Information S5 that the maximum payload ability of the ideal SSR is over 460 N, some 480 times its own weight, just by the two tiny 5 V DC motors. However, the measured maximum stationary suction force without sliding of the SSR in the water-wetted condition was 50.3 N (52 times its own weight), for which the measurement method is given in Supplementary Information S6. This difference is attributed to the non-optimised structure and materials of the fabricated suction cup and suggests an opportunity to improve suction loading through future materials selection and structural design. In our experiments we demonstrated loaded suction sliding with 1 kg (10 times the weight of the robot) which we believe is sufficient to demonstrate its high load-carrying ability. This payload ability is comparable to snails, for which payload ability has been reported as more than 20 times their weight[27,28]. Another unique advantage of the SSR is that the adhesion during sliding suction is constant. In contrast, multi-feet climbing robots will lose $n/m$ ($n$: number of detached adhesive pads, $m$: total number of adhesive pads) adhesive force during the movement.

## Discussions on low energy consumption

According to the calculation in Supplementary Information S7, the maximum power consumption of SSR is 1.7 W (all electronics working at their rated power) and the theoretical working time of SSR is approximately 1 hour (powered by the battery). When the robot is statically adhering to the substrate, the power consumption is 0 W (all the electronics turned off). This is because energy is only required for driving the sliding movement, secreting water and attaching to/detaching from the substrate, and the suction adhesion requires zero energy on smooth surfaces since the leakage on smooth surfaces is neglectable. Based on our test, the SSR stably adheres upside-down to the PMMA surface overnight, which is long enough for practical applications. The inner volume of the hemispheric portable tank is 16.7 cm³ which can supply water for ~619 cm sliding. In summary, the SSR can operate with low energy and water consumption.

## Limitations

The current design of SSR can only slide on relatively smooth surfaces. Surface roughness will reduce the sliding suction performance since it will cause leakage and generate higher friction. However, it will increase the static friction on tyres to avoid slippage. Future designs can use a small additional vacuuming pump to compensate for the leakage during sliding on rough surfaces. A pump-assisted SSR will be much more energy efficient than regular pump-assisted wall-climbing robots (and is predicted to use approximately four orders of magnitude less energy), thanks to the smaller leaking orifice. Detailed discussions are provided in Supplementary Information S9. In addition, the SSR cannot slide over a step-like change on the surface. Future design can use multiple suction cups to move such discrete topographic features. Moreover, sensors can be embedded to enable closed-loop control for precise movement and to autonomously correct for movement deviation caused by tyre slip and transient stiction. Water tubes can be positioned close to the head, analogous to the snail's mucus gland location. This will make wetting of the leading edge more effective, increasing the speed of climbing. Furthermore, methods for increasing the CoF of the tyres (e.g., super hydrophobic coating) can reduce accidental tyre slip when sliding through the water-wetted area.

In this paper, we propose an adhesive locomotion strategy—water-enhanced soft sliding suction—which allows the robot to achieve strong suction adhesion and controllable sliding movement at the same time. The sliding suction is achieved by reducing the friction and enhancing the suction at the suction interface by exuding water to act as both a lubricant and a suction enhancer. We introduced the methods and demonstrated the practical application of this strategy in a sliding suction robot, SSR. The SSR successfully achieved rotational and translational sliding suction on substrates at a wide range of tilt angles, including vertical walls and upside-down ceilings. The SSR also demonstrated high loaded sliding ability, carrying 1 kg mass which is 10 times heavier than itself. This sliding suction mechanism shows an advance over previous adhesive locomotion strategies. It consumes relatively low energy but achieves high payload ability; it allows the robot to move in a continuous sliding motion with constant high adhesive force, which is entirely different from the step-by-step walking motion of the previous climbing mode. Through the performance of SSR, we demonstrated that sliding suction offers low energy consumption, high adhesion efficiency and safety, high loading capacity and low complexity, while only leaving a quick-to-evaporate water trail. These features endow sliding suction with great potential for future applications in robotic fields, including industrial gripping, climbing, outdoor inspections and transportation.

## Methods

### Designing the sliding suction cup

The sliding suction cup is designed with a diameter of 50 mm and a slope angle of 10°. The suction cup disc is made from casting PU rubber (DuroFlex-60, Polytek) which is much stiffer than the bottom silicone pad, to strengthen the whole suction cup. The silicone pad is made from casting silicone (EcoFlex 00-30, Smooth-on), while the $S_{HydPho}$ is with 0% PBP and the $S_{HydPhi}$ is with 2% w/w PBP. The geometry, fabrication and preliminary test of the suction cup are given in Supplementary Information S2, 3 and 6.

### Structure details of the sliding suction robot

The rigid parts of the robot (e.g., the frames, gears, etc.) were 3D printed with VeroGrey (J826, Stratasys). The soft parts (i.e., the snapping membrane and wheel tyres) were made from casting silicone

(Dragon Skin 10NV). A battery (542730, EEMB) is used for powering, and a voltage booster board (DollaTek) is used to increase the voltage from 3.7 V to 5 V to drive pumps and motors. The control unit consists of an ESP32 development board (Wemos D1 Mini, ALMOCN) and two motor drivers (L293D, Youmile). One of the drivers actuates the two DC motors and another actuates the two pumps. The rotating speeds of the motors and pumping speeds of the pumps are controlled by pulse-width modulation (PWM).

## Data availability

All data supporting this work are available at the University of Bristol data repository, data.bris, at https://doi.org/10.5523/bris.1zdz8b5iw6f662pqwqf4kh8r55, or within the Supplementary Information files.

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

## Acknowledgements

T.Y. was funded by the Chinese Scholarship Council through award 201906120027. J.R. was supported through EPSRC research grants EP/V062158/1, EP/T020792/1, EP/V026518/1, EP/S026096/1 and EP/R02961X/1, and by the Royal Academy of Engineering as a Chair in Emerging Technologies, CIET1718/22.

## Author contributions

T.Y. proposed the concept, performed the experimental studies and carried out the analysis. T.Y. and J.R. wrote the manuscript. H.G. and J.R. supervised the work.

## Competing interests

The authors declare no competing interests.
