## [Peer Review File · Nature Communications]

REVIEWER COMMENTS

Reviewer #1 (Remarks to the Author):

General comments: Inspired by the Gastropods that can achieve sliding suction through mucus secretions, this work presents a water-enhanced soft robotic sliding suction mechanism that not only can generate stable adhesion, but also can achieve slide with low friction force. With the sliding suction mechanism, an untethered sliding suction robot is further developed that can achieve high rotational speeds of $52^\circ/\text{s}$ and translational movement speeds of $20.8\text{mm}/\text{s}$. The robot also shows the advantage of high payload ability, low energy consumption, and tracking complex trajectories. In general, the idea and results of this work are interesting. However, the significant revisions should be addressed for further consideration, including, i) the current version looks rough without rigorous theoretical analysis on the sliding force; ii) the structure is confusion with several parts on working principle, Materials and Methods, and tests; iii) the data analysis and applications are too simple to show the advantages of the development with existing works.

Specific comments:

1. In the Introduction, the sliding suction still remains elusive, but the challenge of the sliding suction is not clear. Does it depend on the materials, fabrications or something else?
2. Both the Fig. 1 and Fig. 2A show the working principle of the snails' sliding suction. It is not clear.
3. In Fig. 2D, more details of the structure of the sliding suction are necessary.
4. The adhesion force of the sliding suction may also rely on the roughness of the substrate. It is better to discuss it.
5. The force analysis seems a qualitative description. The data figures are not well characterized with different substrates.
6. This work aims to propose one kind of water-enhanced soft sliding suction for climbing robots. However, a lot of works have been reported for the climbing robots. It is better to systematically analyze the design, fabrication and control of the suction, such as the influence of geometric parameters on the adhesion forces.
7. The application is too simple to show the effectiveness of the development. In this version, only one application has been adopted. It is better to demonstrate more demo to valid the generality of this sliding suction.

Reviewer #2 (Remarks to the Author):

This paper presents a water-enhanced soft sliding suction method, which allows the climbing robot to achieve strong suction adhesion and controllable sliding motion together. I have interestingly read the paper and believe that the paper was well-written and organized. Following are my questions and comments.

1) The weight of the robot was not included in consideration. The authors also mentioned that moments generated by the robot's weight are neglectable. However, The weight of a climbing robot must be a critical factor for analyzing and evaluating its performance. Through the experimental results provided, it is also clear that without considering the robot's weight, the center deviation cannot be analyzed and explained. The phenomena of the robot motion given in the manuscript cannot be well explained in enclosing the weight effect. The reviewer suggests the author re-formulate and analyze the robot's performance again, including the weight factor.

2) The authors noted that "the suction cup adhesion does not require any energy." This seems not true. Any suction cup can't avoid the air leak. Is there no air leak in your suction cup? How do you prevent this air leak?

3) The known facts may not be necessary for the manuscript, such as the explanation of the game controller in lines 272-277 (it does not affect the contents of the paper), and subsection 4.1 "movement deviation." You can use the space to explain the unclear part, like the equation (6) (how do you have it?) , the experimental results shown in figure 5 (why is the movement deviation for the rotation motion in 90 Deg case smaller than 45 Deg and 135 Deg? Why does the robot move sideways even driving it up and down in 0 Deg and 180 Deg?)

4) The accuracy of the user-controlled route tracking must depend on the experience and skill of the driver. It is not clear what performances need to be evaluated.

5) For the S5 "Measurement of the suction cup bending force....", it is valuable to refer paper (Dingxin Ge, Takahiro Matsuno, Yi Sun, Chao Ren, Yongchen Tang, and Shugen Ma, Quantitative Study on the Attachment and Detachment of a Passive Suction Cup, Vacuum, 116, 2015, pp.13/20)

6) Other minor comments:

a) It is understandable that "SSR untethered (Fig .6A)" in line 228

b) You seem to use $g = 10$ for the explanation in line 190. For a scientific paper, $g = 9.8$, at least, should be used.

c) The robot's weight is not considered in equation (1), but there is no explanation.

d) For S6.3 "the Loading ability" means the maximum suction force of the cup, but not pulling the payload up. Please check this.

Reviewer #3 (Remarks to the Author):

Report on "Water-enhanced..." by T. Yue et al.,

I think this is an interesting study but in its present form of limited practical usefulness.

Here are my comments:

(1) The proposed set-up will only work on relative smooth surfaces as otherwise the suction cup will not seal.

(2) To move the "robot" use wheels but the snail can move just using body movements. Would it be possible to change the design to operate more like the snail without external wheels. It would be interesting to know how the snail is able to move.

(3) The authors mention that Gecko adhesives are problematic on contaminated surfaces. But I believe this is the case also for the suction cup mechanism suggested in this study. If the substrate contains e.g. dust and sand particles then the suction cup is unlikely to seal the contact.

(4) Another problem is if there is a step-like change in the surface topography. The sliding suction cup cannot move over a step-like change in the surface height even in the downhill direction because it would not seal when overlapping the step. On the other hand a robot with several suction cups moving step-like like humans could move over a step.

(5) The physics of suction cups has been studied in Ref. "Physics of suction cups", A Tiwari, BNJ Persson, Soft matter 15 (46), 9482-9499 (2019). The author may want to study this reference.

The same authors have also published an article about suction cups in water.

(6) The rubber wheels are made from silicone rubber and the authors state that this result in large friction

also in water due to dewetting. But this is not necessarily the case if the substrate is wetted by water as in that

case the interface will be hydrophilic even if the wheel surface is hydrophobic.

Response Letter to Reviewer Comments

Dear Editor and Reviewers,

The authors would like to thank the efforts of the editorial personnel and the reviewers. We have carefully checked all the reviewers' comments and made corresponding changes in the manuscript. In this response letter, **the reviewers' original comments are marked as red**, **our replies are marked as blue**, and **the changes made in the manuscript are in quotes and marked as purple**. Below we answer the reviewers' comments in detail.

Reviewer #1,

General comments: Inspired by the Gastropods that can achieve sliding suction through mucus secretions, this work presents a water-enhanced soft robotic sliding suction mechanism that not only can generate stable adhesion, but also can achieve slide with low friction force. With the sliding suction mechanism, an untethered sliding suction robot is further developed that can achieve high rotational speeds of 52°/s and translational movement speeds of 20.8mm/s. The robot also shows the advantage of high payload ability, low energy consumption, and tracking complex trajectories. In general, the idea and results of this work are interesting. However, the significant revisions should be addressed for further consideration, including, i) the current version looks rough without rigorous theoretical analysis on the sliding force; ii) the structure is confusion with several parts on working principle, Materials and Methods, and tests; iii) the data analysis and applications are too simple to show the advantages of the development with existing works.

Reply: The authors appreciate your valuable comments and believe they have helped us improve the manuscript.

To address your comment, i) the current version looks rough without rigorous theoretical analysis on the sliding force, we have reorganised the sections **Physical models of the SSR** and **Supplementary Information S6 Detailed force analysis and calculations** and expanded formulae to make the force analysis more rigorous. Please note that, to save space in the manuscript, most of the detailed force analysis and formula deviations were provided in **Supplementary Information S6**, and the **main text** only includes necessary governing equations.

To address your concern, ii) the structure is confusion with several parts on working principle, Materials and Methods, and tests, we apologise that the first submission did not fully comply with Nature Communication's manuscript structure. In the revised version, we have reformatted the manuscript structure as Introduction – Results – Discussion – Methods. We also have added two new sections – **Supplementary Information S3 Design and structural details of SSR** and **S4 Fabrications and assembling methods of SSR** to provide clearer structure, materials and methods for readers to easily follow.

To address your concern, iii) the data analysis and applications are too simple to show the advantages of the development with existing works, we have implemented more experiments, including more characterization experiments of the robot sliding suction performance and two new application experiments, and have made a deeper analysis of the sliding suction mechanism in the **Discussion**. Please see our replies to your specific comments (below) in detail.

Specific comment #1: In the Introduction, the sliding suction still remains elusive, but the challenge of

the sliding suction is not clear. Does it depend on the materials, fabrications or something else?

Reply: Thank you for the comment. Although the sliding suction has not been widely reported by previous studies, it is a common daily phenomenon. For example, when we put a glass cup on a table wetted by a thin water film, we may feel that the cup is difficult to pick up (generating a strong suction) but can easily slide on the table. This is an example of sliding suction. In other words, sliding suction is an adhesion strategy different from most previous adhesion strategies – it creates a strong adhesive force but does not affect the relative sliding movement between the two contacting surfaces. Our goal is to understand the underlying principle of this phenomenon and apply it to soft robotics. The challenge of sliding suction is to address the conflict between the sliding movement and the suction. For example, a rubber suction cup will eventually de-adhere as it is pulled along a flat substrate. Here we take inspiration from the similarity between snails and the “sliding cup”, and find that a liquid-enhanced seal is critical. To create a liquid seal between the suction interface for robotic use, the challenge involves both the materials and structure design. For clarity, we have made the following changes in the **Introduction** (page 2, line 38) of the revised manuscript:

Edit: “This contradicts the stereotype that a regular suction cup is difficult to slide on the substrate and is susceptible to falling off when applied by a tangential force. However, a common phenomenon – a cup can easily slide on a wet table with strong suction – indicates that the suction and sliding movement of suction cups are not a conflict. The liquid film at the suction interface is the common feature found between snails (i.e., mucus) and the “sliding cup” (i.e., water). As reported by several studies, a thin liquid film could reduce the friction between two contacting surfaces¹ and enhance the suction². Therefore, we believe that the critical challenge for designing a snail-like sliding suction cup involves materials and structure design for intelligently introducing the liquid film into the suction interface while maintaining a compliant and effective suction.”

Specific comment #2: Both the Fig. 1 and Fig. 2A show the working principle of the snails’ sliding suction. It is not clear.

Reply 1: Thank you for this comment. For clarity, we have made the following changes: 1. We have removed Fig. 1 and redundant text in the **Introduction**. 2. We have added more explanations in the **Principle of sliding suction mechanism** (page 3 line 74).

Edit 1: “Snails secrete mucus to lubricate the contacting interface and reduce the coefficient of friction (CoF), making sliding movement possible. In addition, mucus can enhance the suction by sealing the gaps between the abdomen and the substrate³. Coordinated body movements, in the form of travelling waves, then generate the driving force to slide forward. Other low-viscous liquids, such as water, also reduce

friction, which previous studies in contact and friction mechanics have well reported¹.”

Reply 2: 3. We have modified Fig. 1A (previous Fig. 2A) by adding an additional diagram to illustrate how the snail can achieve movement via the muscular travelling pedal wave (as marked below).

Edit 2:

Fig. 1. The sliding suction mechanism. (A) The snail achieves sliding suction by mucus secretion and muscular contraction. IW: interwave. PW: pedal wave. ... (D) The experimental setup for evaluating sliding suction. ... Insert: difference of wetting between the hydrophilic and hydrophobic silicone pads submerged in water. ... (F) Measured kinetic friction force as a function of the perpendicular pulling force of the S_{HydPhi} -water case on different substrates. Kinetic friction force corresponds to the flat period denoted by $f_{kinetic}$ in (E).

Specific comment #3: In Fig. 2D, more details of the structure of the sliding suction are necessary.

Reply: Thank you for this comment. Fig. 2D (now Fig. 1D) is the experimental setup to measure the kinetic friction force during the sliding suction. To clarify, we have added an insert in Fig. 1D (please refer to the Fig. 1D above) to show more details of the sliding suction. As the insert shows, when the surface of the suction cup bottom is hydrophilic, water is drawn into the gap by capillary force; therefore, the van der Waals force (i.e., friction) are significantly reduced. When the surface is hydrophobic, air remains in the gap, permitting the development of van der Waals forces which generate friction. This explains why the sliding suction is achieved.

Specific comment #4: The adhesion force of the sliding suction may also rely on the roughness of the

substrate. It is better to discuss it.

Reply 1: Thank you for this comment. We agree that roughness will affect the sliding suction performance from two aspects: for suction, roughness will cause leakage, which causes the suction failure; for sliding, a rough surface might generate higher friction force, reducing the sliding movement in comparison to a smooth surface; however, roughness might also enhance the static friction between wheel tyres and the surface and thereby reduce wheel slippage. In general, rough surfaces are considered more challenging for the SSR to achieve sliding suction.

To better understand the influence of surface roughness on sliding suction performance of the SSR, we have performed new experiments and added new discussions. We engraved PMMA sheets with thin grooves with different depth and let the SSR slide on these surfaces. Suction pressure was measured during the sliding. Experimental results show that a higher surface roughness reduces the time that the SSR can slide on the substrate. In addition, the sliding movement also increases the leakage rate by approximately half. However, this does not mean that the sliding suction cannot be applied to rough surfaces, rather that it is less effective and additional mitigation can be employed to maintain suction, for example by adding a small supplementary pump: we have theoretically demonstrated that the pump-assisted SSR consumes much lesser ($O(10^{-4})$) energy than traditional pump-assisted wall-climbing robots. The experimental results on rough surfaces and theoretical analysis have been provided in a new section **Supplementary Information S9 Discussion of the pump-assisted design for sliding on rough surfaces** (page 7, line 186) as below.

Edit 1: “S9 Discussion of the pump-assisted design for sliding on rough surfaces

To determine the influence of surface roughness on sliding suction performance, we conducted a series of experiments on rough surfaces. Three rough surfaces were prepared by engraving flat PMMA sheets. We use a FusionEdge (EpilogLaser) laser cutter to engrave parallel lines (0.5 mm distance) with different depths. From surface 1 (denoted as S1, smoothest) to surface 3 (denoted as S3, roughest), we set the engraving laser parameters as 100 % speed, 5 / 10 / 15 % power and 5000 Hz. The engraved surfaces are shown in Supplementary Fig. 7A. To measure the pressure inside the suction region, we drilled a thorough-hole (1 mm diameter) on the bottom centre of the suction cup, and connected the hole with thin silicone tubes to a pressure sensor (SSCDRRV015PDAA5, Honeywell). Data was read by an Arduino Nano board. Experiments were conducted via following steps: First, we wet the rough surface with a thin film of water. Second, we manually squeezed the SSR to the surface then released it, from which the suction is deemed to begin (as the black dashed line shows in Supplementary Fig. 7B). We separate the suction into two states: static and sliding. For static suction, we did not activate the movement of SSR; for the sliding

suction, we activate the forward sliding movement parallel to the grooves at the moment of release (black dashed line). According to Supplementary Fig. 7B, surface roughness reduces both the maximum suction pressure and the suction longevity. In addition, the sliding movement does not affect the maximum suction pressure on rough surfaces compared with the static state; however, the suction longevity is reduced by approximately 50 %. During the sliding, new percolation channels between the environment and the suction region are introduced by the relative movement of the two contacting surfaces. We attribute the longevity reduction caused by the sliding movement to this dynamic change in leakage channels. In contrast, the percolation channels of a static suction cup are relatively stable and constant, and their number and form only change by leakage-induced shape recovery of the suction cup. We will not analyse more on the micro contact mechanics⁴⁻⁶ since this is not the focus of this paper; however, given the experimental results in Supplementary Fig. 7, we can draw the conclusion that the sliding-induced leakage is in the same order of magnitude as the roughness-induced leakage.

Supplementary Fig. 7. Sliding suction test on rough surfaces. (A) Rough grooved surfaces prepared by laser engraving. Scale bar: 1 mm. (B) Measured suction pressure on rough surfaces in static and sliding suction situations.

However, this does not mean that the sliding suction cannot be applied to rough surfaces. Additional mitigation can be employed to maintain suction; for example, a small vacuum pump can be connected to the suction region to compensate for leakage. This design is similar to traditional wall-climbing robots^{7,8}, however, the SSR will consume much less energy according to our analysis below. Previous pump-assisted wall-climbing robots require the body to be lifted 1-5 mm (typically 5 mm for commercial products⁹) from the substrate to avoid direct contact and friction; however, this causes severe leakage and accordingly needs a strong vacuuming flow. In contrast, the sliding suction method employs direct body contact with the substrate. The direct contact with a rough surface (e.g., coarse sandpaper with grit >40) leaves a mean gap of $\sim 200 \mu\text{m}$ (we use half of the granule size of the 40-grit sandpaper for estimation)¹⁰. According to

the leakage model¹¹, the leakage rate Q of a leaking suction cup can be estimated by

$$Q = \frac{L_y d^3 (p_a - p_b)}{12\eta L_x},$$

where L_x and L_y are the width and length of the sealed region of the suction cup, p_a and p_b are the pressure of the outer environment and the internal suction region, d is the gap width (for the pump-assisted wall-climbing robot, we use $d = 5$ mm; for the pump-assisted sliding suction robot, we use $d = 200$ μ m), η is the viscosity of the fluid (for the pump-assisted wall-climbing robot, the fluid is air so $\eta = 5 \times 10^{-5}$ Pa·s; for the pump-assisted sliding suction robot, the fluid is water so $\eta = 1 \times 10^{-3}$ Pa·s). Assuming they both require the same pressure differential to adhere to the substrate (typically $p_a - p_b = 2.5 \sim 4.5$ kPa^{7,8}) and have the same size (i.e., same L_y/L_x), a simple calculation can derive the ratio of the flow rate:

$$\frac{Q_{\text{centrifugal}}}{Q_{\text{sliding_suction}}} = \frac{d_{\text{centrifugal}}^3 \eta_{\text{water}}}{d_{\text{sliding_suction}}^3 \eta_{\text{air}}} = 3.125 \times 10^5$$

The ratio of the power consumption for maintaining the suction is:

$$\frac{P_{\text{centrifugal}}}{P_{\text{sliding_suction}}} = \frac{Q_{\text{centrifugal}} d_{\text{centrifugal}} \rho_{\text{air}}}{Q_{\text{sliding_suction}} d_{\text{sliding_suction}} \rho_{\text{water}}} \approx 1.01 \times 10^4.$$

This indicates that the power consumption on a rough surface (~ 40 grit) of a pump-assisted sliding suction robot is four orders of magnitude less than a regular pump-assisted wall-climbing robot, demonstrating a significant advance. In addition, we also consider comparing the pump-assisted sliding suction robot with wall-climbing robots based on the Bernoulli effect, which uses a reverse high-speed airflow. A Bernoulli-effect-based climbing robot requires a much higher flow rate to generate the pressure drop than suction-based robots, which consumes significantly more energy¹².”

Reply 2: In the **main text** of the revised version, we have also added following text in the **Discussion** (page 14, line 400).

Edit 2: “The current design of SSR can only slide on relatively smooth surfaces. Surface roughness will reduce the sliding suction performance since it will cause leakage and generate higher friction. However, it will increase the static friction on tyres to avoid slippage. Future designs can use a small additional vacuuming pump to compensate for the leakage during sliding on rough surfaces. A pump-assisted SSR will be much more energy efficient than regular pump-assisted wall-climbing robots (and is predicted to use approximately four orders of magnitude less energy), thanks to the smaller leaking orifice. Detailed discussions are provided in Supplementary Information S9.” ...

Specific comment #5: The force analysis seems a qualitative description. The data figures are not well characterized with different substrates.

Reply 1: Thank you for this comment. In the previous version, we provided critical qualitative equations in the force analysis in the **main text**. The reviewer may have missed the analysis details and quantitative calculations in **Supplementary Information S5** (page 4, line 95). We agree that our previous force analysis in the **main text** is not well organised for readers to follow. In the revised version, we have reorganised the force analysis and added the calculation results to the section **Physical models of the SSR** (page 8, line 213), which we believe is easier for the reader to follow. The leakage model of the suction mentioned in the **above reply (Edit 1) to Specific comment #4** can further improve the physical model of sliding suction; however, since the leakage analysis is not the focus of this work, we provide it in the **Supplementary Information**.

Edit 1 (qualitative analysis in main text):

“Physical models of the SSR. Here, we provide critical formulae in this section and put detailed derivations in Supplementary Information S6. Key parameters are the sum of the static friction force on two tyres, f_{tyre} , and the kinetic friction force on the suction interface, f_{kinetic} . f_{tyre} and f_{kinetic} are in the same plane (defined as the “sliding plane”, as shown in Fig. 2E). We assume the SSR is sliding with constant velocity V on a flat substrate (tilt angle α). The influence of tilting moment (i.e., the mass of the robot times the distance of centre-of-mass from the substrate) is neglectable. This is because it affects the contacting pressure distribution but not the total contacting force at the suction interface, while we assume the CoF is constant and not affected by the pressure change; in addition, the robot is lightweight (96 g), has low centre-of-mass (~20 mm) and a relatively large suction cup (diameter of 50 mm), making the influence of tilting moment further neglectable. f_{tyre} can be derived by the tangential force balance,

$$\vec{f}_{\text{tyre}} + \vec{f}_{\text{kinetic}} + m\vec{g} \cdot \sin\alpha = 0, \quad (2)$$

where $m = 0.096$ kg is the mass of the SSR. Now f_{kinetic} is the only unknown parameter in the model, which can be calculated via $f_{\text{kinetic}} = \hat{\mu}_{\text{wat_suc}} F_{\text{bend}}$ in Eq. (1). Through Fig. 1F we find that f_{kinetic} is influenced by F_{pull} ; therefore, we know F_{bend} is affected by F_{pull} since $\hat{\mu}_{\text{wat_suc}}$ is constant. F_{bend} , the bending-induced force when a suction cup is adhering to the substrate, can be measured by steps described in Supplementary Information S5. We measured $F_{\text{bend}} = 1.6$ N when $F_{\text{pull}} = 0$ N. According to the result in Fig. 1F,

$$\hat{\mu}_{\text{wat_suc}} = \frac{f_{\text{kinetic}}(F_{\text{pull}}=0)}{F_{\text{bend}}(F_{\text{pull}}=0)} = 0.29 = \text{const.} \quad (3)$$

We observe that the calculated normalised CoF of the sliding suction cup $\hat{\mu}_{\text{wat_suc}} = 0.29$ is greater than the measured CoF of the flat silicone pad $\mu_{\text{wat_phi}} = 0.1$ (Fig. 1C), although they employ the same

contacting materials – hydrophilic silicone and PMMA – in the same water-contact condition. The relation between $\hat{\mu}_{\text{wat_suc}}$ and $\mu_{\text{wat_phi}}$ is

$$\hat{\mu}_{\text{wat_suc}} = \frac{f_{\text{kinetic}}}{F_{\text{contact}}} = \frac{\iint \mu_{\text{wat_phi}}(s)p(s) ds}{F_{\text{bend}}}, \quad (4)$$

where p is the contacting pressure and S is the contacting region between the suction disc and the substrate. Several reasons cause the difference: First, the suction cup’s silicone pad was cured in closed moulds, while the flat silicone pad was cured in open moulds; therefore, the suction cup’s bottom surface is rougher than the flat silicone pad. Second, the non-flat geometry of the suction cup influences the surface micro-topography and therefore air is not completely squeezed out of the suction interface during the adhesion, which generates unwetted sites and adds static friction force. The measured normalised CoF on hydrophobic wheel tyres in the water-contact case shows the same trend: $\hat{\mu}_{\text{wat_tyr}} = 5.56 > \mu_{\text{wat_pho}} = 1.23$. However, just as $\mu_{\text{wat_phi}} \ll \mu_{\text{wat_pho}}$, $\hat{\mu}_{\text{wat_suc}} \ll \hat{\mu}_{\text{wat_tyr}}$ enables sufficient wheel grip to initiate and maintain sliding suction.

According to $f_{\text{kinetic}} = \hat{\mu}_{\text{wat_suc}}F_{\text{bend}}$ in Eq. (1), once we obtain $\hat{\mu}_{\text{wat_suc}}$ and the $f_{\text{kinetic}} - F_{\text{pull}}$ relation (Fig. 1F), the $F_{\text{bend}} - F_{\text{pull}}$ relation can be calculated as

$$F_{\text{bend}} = 0.12F_{\text{pull}} + 1.54. \quad (5)$$

Eq. (5) indicates that the increase of F_{pull} slightly increases F_{bend} ; therefore, f_{kinetic} is further slightly influenced by F_{pull} . The decrease of the internal pressure p_{in} , which does not consume energy, balances most of the increase of F_{pull} , according to $F_{\text{pull}} = (p_{\text{atm}} - p_{\text{in}})A - F_{\text{bend}}$ in Eq. (1). Although the decrease of p_{in} makes the suction cup squeeze harder, it does not change the geometry very much; this is why F_{bend} only increases slightly and the sliding suction mechanism has such high loading ability. Eq. (1), (2), (3) and (5) are the critical formulae of the physical model of the SSR, through which we calculated the sliding ability, climbing ability and loading ability of the SSR. Details of the formula derivation are provided in Supplementary Materials S6. Calculation results suggest that the SSR can easily slide upside-down, climb on the wall and carry a heavy mass.”

Edit 2 (quantitative calculation in Supplementary Information):

“S6 Detailed force analysis and calculations

S6.1 Sliding ability

The “sliding ability” we refer to is the ability of the SSR to slide on the substrate. Since gravity acting on the SSR and the pulling force may add force components at the suction interface, we calculate the sliding

ability in the simplest condition. No external pulling force is exerted on the SSR and the SSR is upside-down adhering to the PMMA ceiling (no tangential gravity component).

The equivalent pulling force $F_{\text{pull_equ}}$ is

$$F_{\text{pull_equ}} = 6F_{\text{spring}} + mg = 6kd_{\text{spring}} + mg,$$

where $k = 0.15$ N/mm is the spring constant of each spring, $d_{\text{spring}} = 3.5$ mm is the compression distance and $mg = 0.94$ N is the gravitational force on the SSR. The kinetic friction force to overcome is

$$f_{\text{kinetic}} = \hat{\mu}_{\text{wat_suc}} F_{\text{bend}}(F_{\text{pull_equ}}) = 0.59 \text{ N}.$$

In the meantime, the maximum static friction force that two wheels can supply is

$$f_{\text{tyre}} = \hat{\mu}_{\text{wat_tyr}} \cdot 6F_{\text{spring}} = \hat{\mu}_{\text{wat_tyr}} \cdot 6kd_{\text{spring}} = 17.51 \text{ N}.$$

The driving force of the SSR comes from two DC motors. When supplied with 5 V, the output motor torque is $\tau_{\text{motor}} = 37.2$ mN·m. The torque is transited through the gears with gear ratio $\eta = 11/28$ to two wheels. Therefore, the total maximal driving force on the wheels is

$$F_{\text{drive}} = \frac{2 \cdot 0.8 \cdot \tau_{\text{motor}}}{\eta r_{\text{wheel}}} = 16.834 \text{ N},$$

where 0.8 is the estimated torque transition efficiency in gears, $r_{\text{wheel}} = 9$ mm is the radius of the wheel including tyre. Based on the former calculations, we know that $f_{\text{kinetic}} < F_{\text{drive}} < f_{\text{tyre}}$, indicating that the kinetic friction force at the suction interface can be easily overcome by the driving force, while they are both less than the static friction force on the tyres (no tyre slip). Therefore, the SSR can easily slide upside-down on the PMMA ceiling.

S6.2 Climbing ability

Here the ‘‘climbing ability’’ we refer to is the ability of the SSR to vertically climb upward on the wall, which is the most difficult condition since the total gravitational force from the mass of the SSR acts in direct opposition to upward climbing. The equivalent pulling force applied on the SSR becomes

$$F_{\text{pull_equ}} = 6F_{\text{spring}}.$$

The total tangential force that needs to be overcome is the sum of the kinetic friction force at the suction interface and the gravity,

$$f_{\text{kinetic}} + mg = \hat{\mu}_{\text{wat_suc}} F_{\text{bend}}(F_{\text{pull_equ}}) = 1.52 \text{ N}.$$

Again, we know that $(f_{\text{kinetic}} + mg) < F_{\text{drive}} < f_{\text{tyre}}$, therefore SSR can easily climb upward on the PMMA wall.

S6.3 Payload ability

Here the “payload ability” is the total pulling force that the SSR can maintain suction against when sliding upside-down on the ceiling (the simplest case). We consider a payload mass m_{load} applied perpendicular to the SSR when it is sliding upside-down on the PMMA sheet. The equivalent pulling force becomes

$$F_{\text{pull_equ}} = 6F_{\text{spring}} + mg + m_{\text{load}}g.$$

Therefore, increasing the payload mass causes f_{kinetic} to increase. Let the kinetic friction force equal the driving force when the two motors are operating at their rated power,

$$f_{\text{kinetic}} = \hat{\mu}_{\text{wat_suc}}F_{\text{bend}}(F_{\text{pull_equ}}) = F_{\text{drive}} = 16.83 \text{ N}.$$

Solving the equation we obtain $m_{\text{load_max}} = 46.67 \text{ kg}$ (an equivalent payload force of $> 460 \text{ N}$), the maximum mass that an ideal SSR can carry. In experiments, we measured a maximum stationary suction force for the SSR of 50.3 N (5.03 kg). This also defines the practical upper bound for loaded sliding using the fabricated suction cup in the SSR. We limited our loaded sliding tests to 1 kg to avoid damaging the robot, which is mainly 3D-printed, and the suction cup, which is non-optimised. The calculated result (above) shows the extraordinary payload potential of SSR, even though it is only driven by two small 5 V DC motors.”

Reply 2: For your comment “The data figures are not well characterized with different substrates”, we suspect you might indicate 1. We did not perform sliding suction experiment on substrates with different materials, or 2. The characterization experiments on tilted PMMA sheets is too simple. For 1., we have performed sliding suction experiments (page 5, line 147, please refer to Fig. 1F in the above **reply to Specific comment #2**) on three different materials – POM, PMMA and aluminium alloy, which have increasing hydrophilicity. Results show that the substrate with higher hydrophilicity has lower kinetic friction during the sliding suction.

Edit 3: “We performed the experiments on substrates made from three different materials with increasing hydrophilicity: polyoxymethylene (POM, water contact angle (WCA) = 76.8°), PMMA (WCA = 68°) and aluminium alloy (WCA = 57°). Results are shown in Fig. 1F. Increasing the pulling force from 0 to 10 N only slightly increases f_{kinetic} on all three substrates. In particular, the f_{kinetic} on POM is the largest and on aluminium alloy is the smallest. This relation is inverse to the hydrophilicity of the three materials, which will be discussed later. In the rest of the paper, we will use PMMA as the default substrate material if not otherwise specified.”

Reply 2: For 2., we have performed more experiments of the sliding suction characterization on tilted PMMA surfaces, as shown in Fig. 3C and F.

Edit 4: “The SSR was actuated to rotate clockwise for 5 seconds and repeat for 3 times on the tilted PMMA sheet. ... The SSR was actuated to translate for 2 seconds and repeat for 3 times on the tilted PMMA sheet. ...”

Fig. 3. Experimental characterization of the sliding suction ability of the SSR. ... (C) d_{cen} and ω_{rot} of the rotation tests. Dashed lines are the average of corresponding parameters. Shadows are the standard deviation. ... (F) V_{trans} and d_{off} of the translation tests.

Specific comment #6: This work aims to propose one kind of water-enhanced soft sliding suction for climbing robots. However, a lot of works have been reported for the climbing robots. It is better to systematically analyze the design, fabrication and control of the suction, such as the influence of geometric

parameters on the adhesion forces.

Reply 1: Thank you for this comment. We agree that lots of work of climbing robots have been reported. As we illustrated in the revised **Introduction**, the main climbing robot technologies include suction^{13,14}, electromagnetic adhesion^{15,16}, gecko adhesion^{13,17} and electro-adhesion^{13,18}. However, most of these technologies have conflicts between adhesion and movement, therefore previous climbing robots must incorporate multiple adhesive pads and have difficulties maintaining a high and constant adhesive force during the movement^{15,19-22}. In this work, we obtain inspiration from snails' sliding suction movement and propose robotic sliding suction mechanism which for the first time achieves a low-energy-consumption, constant and strong adhesive locomotion compared to previous work. The proposed sliding suction mechanism is an entirely new technology different from previous reported wall-climbing robotic technologies. In the revised manuscript, we have re-illustrated the issues of current climbing strategies in **Introduction** (page 1, line 23).

Edit 1: "Climbing robots have been developed with various adhesion strategies, including suction^{13,14}, electromagnetic adhesion^{15,16}, gecko adhesion^{13,17} and electro-adhesion^{13,18}. The typical target for these climbing robots is maximising the adhesive force and, hence, the holding or perching security of the robot. However, the higher the adhesive force generated, the higher the induced friction force between the adhesive pad and the substrate. To operate under the restriction of a high friction force, climbing robots must have multiple adhesive pads and intermittently detach, move and reattach the pads to climb, resulting in a discrete "walking" movement^{15,19-22}. This discrete gait has several shortcomings: the control system must frequently break and regenerate the adhesive force on each pad; a complex transmission structure must be built into the robot, adding weight; the repositioning of the adhesive pad and the regeneration of the adhesive force waste energy; and the total adhesive force is significantly lower during movement since some of the adhesive pads are detached."

Reply 2: We also have stressed again the difference and advances of the proposed sliding suction mechanism to previous climbing robots in **Discussion** (page 14, line 422).

Edit 2: "This novel sliding suction mechanism shows a significant advance over previous adhesive locomotion strategies. It consumes relatively low energy but achieves high payload ability; it allows the robot to move in a continuous sliding motion with constant high adhesive force, which is entirely different from the step-by-step walking motion of the previous climbing mode."

Reply 3: For clarity, we have reorganised the manuscript structure to: Introduction – Results – Discussion – Methods. Please see changes made in the revised manuscript **main text**. In the revised manuscript, we provide more information on the design and fabrication of the proposed sliding suction robot, and have

added the following two sections **Supplementary Information S2 Design and structural details of SSR** (page 2, line 20) and **S4 Fabrications and assembling methods of SSR** (page 4, line 81) to provide structure, materials and methods details for readers to easily follow.

Edit 3:

“S3 Design and structural details of SSR

The design of SSR begins with the suction cup. The fundamental structure of the suction cup, as we describe in the main text, is a double-layer structure composed of a top PU reinforcement layer and a bottom PBP-silicone pad. The diameter of the suction cup is selected as 50 mm which is a suitable size for practical demonstrations. The key geometrical parameter of the suction cup design is the slope angle, as shown in Supplementary Fig. 2. We tested three slope angles with 10°, 20° and 30° and no difference was observed between the sliding suction performance of three suction cups, while the 10° suction cup generates a slightly larger maximum suction force than the other two. It can be explained by Equation (1) in the main text, $F_{\text{pull}} = (p_{\text{atm}} - p_{\text{in}})A - F_{\text{bend}} + mg$. Larger slope angle makes the suction cup generate larger bending force against the substrate, therefore the maximum suction force is accordingly reduced. Considering the application scenario of the current design is sliding on flat and smooth surfaces, 10° is considered the optimal slope angle.

Supplementary Fig. 2. Geometrical dimensions of the suction cup structure.

The second important design is to control the attachment and detachment of the suction cup. The action of the suction cup attaching to the substrate builds a new seal and detachment from the substrate requires breaking of this seal. A mechanism should be therefore needed to place and lift the suction cup rim to build and break the seal respectively. We use a snapping membrane to achieve this, adopting a similar mechanism as proposed in our previous work^{14,23} and shown in Supplementary Fig. 3. An air chamber is composed of a rigid upper lid (6 in Supplementary Fig. 4) and a soft snapping membrane (5 in Supplementary Fig. 4). On the upper lid there is a thin through-hole with 0.5 mm diameter. The snapping membrane is monostable: when the air chamber holds a negative pressure (vacuumed by the pump, $\sim <20$ kPa), the toroidal-shaped snapping membrane snaps backward; therefore, this backward contraction lifts the suction cup rim (i.e., detaching). Due to the mono-stability, when the air pump stops vacuuming, air

leaks into the chamber through the hole on the upper lid and the snapping membrane snaps forward to place down the suction cup rim (i.e., attaching). This design allows the SSR to use just one tiny air pump to control the states of the suction cup, avoiding the need for a complex circuit.

Supplementary Fig. 3. The activation of the snapping chamber via a small diaphragm pump. (A) The pneumatic circuit for activating the snapping chamber. (B) Left: side view during the activation. Right: perspective view.

The third critical design is the driving system for achieving sliding movement. As shown in Fig. 2 in the main text, we use two wheels to generate sliding movement. Although the sliding suction cup generates a very small kinetic friction force (Fig. 1E and F) during the sliding, we still require the wheels to generate sufficient static friction force, depending on the situation, e.g., to overcome the gravity during vertical climbing. Based on the analysis of the perpendicular pulling payload force on the kinetic friction in the main text, we know that pulling the suction cup does not significantly increase the kinetic friction. However, pushing the wheels onto the substrate, of which the tyre is made from hydrophobic silicone (Dragon Skin 10 NV), significantly increases their static friction and grip. Therefore, we can utilise the antagonistic force between the suction cup and wheels to increase the wheels' static friction. We use a circular array by six compressive springs (7 in Supplementary Fig. 4) to generate the antagonistic force between the suction cup and wheels. The selection of the springs' parameter is determined considering appropriate force and size. After that, DC motors and gears were designed to drive the movement.

The final design is the control system. We use a Wemos D1 mini board to receive controlling signals from a PS4 game controller, and two L293D motor drivers to respectively control the two DC motors and two pumps.

The detailed structure of SSR is shown in the exploded diagram in Supplementary Fig. 4. Explanations of each component are provided in Tab. S1.

Supplementary Fig. 4. The detailed structure of the SSR. 1~24 components are explained in Tab. S1.

Silicone tubes (for air and water transportation) and wires are not shown.

Tab.S1 Information of key components and electronics.

No.	Name	Specifications	Function
1	SA foam	Thickness 2 mm	Water reservoir
2	PBP-silicone	Thickness 1 mm, $\phi 50$ mm	Generate capillary force to water
3	PU reinforcement layer	Thickness 1 mm, $\phi 50$ mm	Strengthen the suction cup

4	Constraining plate	Thickness 1 mm, ϕ 50 mm	Improve the mono-stability of snapping membrane
5	Snapping membrane	Thickness 1 mm	Lift/put down the suction cup rim
6	Upper lid	With a ϕ 0.5 mm hole	Connecting; leaking for putting down the suction cup rim
7	6 compressive springs	5×0.3×10 mm, 0.15 N/mm	Enhance friction on wheels
8	Supporting piece 1	N/A	Supporting
9	Chassis	N/A	Supporting
10	Supporting plate 1	N/A	Supporting
11	Printed circuit board	N/A	N/A
12	MCU	Wemos d1mini 32	Computing
13	Motor driver	L293D	Control motors
14	Supporting plate 2	N/A	Supporting
15	Supporting piece 2	N/A	Supporting
16	Water tank	3D printed by SLA printer	Reserve water
17	Diaphragm Pump 1	3.7~6 V	Supply water
18	Battery	3.7 V, 340 mAh	Power supply
19	Voltage booster	3.7 V to 5 V	Regulate voltage
20	Diaphragm Pump 2	3.7~6 V	Activate snapping chamber
21	DC motor	2.5~5 V, max 37.2 mN · m	Movement
22	Gears	11:18:28 (top to bottom)	Transmission
23	Wheel with silicone tyre	ϕ 18 mm	Generate static friction
24	Supporting piece 3	N/A	N/A

S4 Fabrications and assembling methods of SSR

We first prepared all components as shown in Supplementary Fig. 5, then assembled the SSR via following steps as shown in Supplementary Fig. 6:

- Step 1: We soldered electric components, assembled them onto the chassis of the SSR and sealed electric components with silicone rubber (Ecoflex 00-30). We assembled the upper lid and snapping membrane with instant glue (Loctite Precision Max).

- Step 2: We localised electric components onto supporting plates on the chassis, and assembled gears and wheels onto the chassis. We assembled the suction cup (composed of the PU reinforcement layer, PBP-silicone pad, SA foam and tubes) beneath the snapping chamber via soft glue (Sil-poxy, Smooth-on).
- Step 3: We assembled the suction unit with the chassis via instant glue.
- Step 4: We connected all tubes to pumps and assembled the water tank to the back of the SSR.”

Supplementary Fig. 5. Components for assembling the SSR.

Supplementary Fig. 6. The assembling steps of the SSR.

Edit 4: In response to your suggestion “such as the influence of geometric parameters on the adhesion forces”, the following explanations have been added in **main text** (page 5, line 123).

Edit 4: “Based on the proposed water self-spreading mechanism, a sliding suction cup as shown in Supplementary Fig. 2 can be designed. The sliding suction cup has a top layer (i.e., the polyurethane (PU) reinforcement layer) and a bottom layer (i.e., the PBP-silicone pad). According to our tests presented in Supplementary Information S3, slope angle of 10° and diameter of 50 mm are the optimal geometry to generate sliding suction.”

Specific comment #7: The application is too simple to show the effectiveness of the development. In this version, only one application has been adopted. It is better to demonstrate more demo to valid the generality of this sliding suction.

Reply: Thank you for this comment. To demonstrate more practical applications of the sliding suction robot, we have implemented two additional demonstrations (page 11, line 318). In the first demonstration,

we recruited three volunteers with no knowledge about the sliding suction mechanism. They were asked to use the PS4 game controller to drive the SSR (with 200 g payload) slide upside-down the PMMA sheet, from the starting point to the destination, avoid obstacles. This application demonstrates that the proposed SSR is easy to use, and has potential for delivering goods in industry. In the second demonstration, we fitted a mop to the SSR. The SSR can carry the mop to erase markers on an upright wall and a ceiling. This application demonstrates that the SSR has potential to be used as a cleaning robot, and also can carry different tools to undertake multiple tasks on surfaces which are difficult for human to access. Together with previous two demonstrations (wall-climbing and upside-down route tracking), we believe this is sufficient to demonstrate the practical applications of the sliding suction robot.

Edit: “Demonstration: user-controlled weight transportation with obstacle avoidance. We recruited three volunteers without knowledge of sliding suction. Volunteers were taught the functions of the buttons on the game controller, and were required to use the game controller to drive the SSR (loaded with 200 g mass) and avoid obstacles from the starting point to the destination. The diagram of experimental setup is shown in Fig. 4B i. All volunteers successfully and smoothly completed the task, as shown in Fig. 4B ii-iv and Supplementary Movie 4.

...

Demonstration: erasing marker on different surfaces. A mop (fabricated by attaching a sponge to a 3D-printed bracket, as shown in the insert of Fig. 4D i) was attached to the SSR. The mop was soaked with isopropanol. Two surfaces, an upright painted metal wall and a PMMA ceiling, were drawn with a cross using a permanent marker pen. The SSR was remotely controlled to carry the mop to approach the permanent marker and then erase it using the mop. The SSR successfully completed the task on both the upright wall and ceiling, as shown in Fig. 4D and Supplementary Movie 4.”

Fig. 4. Demonstration of applications of SSR. ... (B) User-controlled upside-down sliding with 200 g payload and obstacle avoidance. i: diagram of the experimental setup. ii to iv: three volunteers successfully controlled the SSR to slide from the starting point to the destination. ... (D) Demonstration of using the SSR as a carrier to clean markers on an upright painted metal wall (i) and an upside-down PMMA ceiling (ii).

Reviewer #2,

General comment: This paper presents a water-enhanced soft sliding suction method, which allows the climbing robot to achieve strong suction adhesion and controllable sliding motion together. I have interestingly read the paper and believe that the paper was well-written and organized. Following are my questions and comments.

Reply: The authors appreciate your positive comments. We believe your suggestions have helped us to improve the quality of the manuscript. Based on your comments, the major changes are:

1. We have considered the weight of the robot, and added more discussions.
2. We have added more discussions on the air leakage. In addition, we have implemented sliding suction tests on rough surfaces.
3. We have implemented more experiments, including more characterization of the robot's sliding suction performance and two demonstrations of practical applications.
4. We have made corresponding changes in response to your detailed comments below.

Major comment #1: The weight of the robot was not included in consideration. The authors also mentioned that moments generated by the robot's weight are neglectable. However, The weight of a climbing robot must be a critical factor for analyzing and evaluating its performance. Through the experimental results provided, it is also clear that without considering the robot's weight, the center deviation cannot be analyzed and explained. The phenomena of the robot motion given in the manuscript cannot be well explained in enclosing the weight effect. The reviewer suggests the author re-formulate and analyze the robot's performance again, including the weight factor.

Reply 1: We appreciate your suggestion for taking the robot's weight into consideration and agree that the weight of the robot should be included in Equation (1) and the previous force analysis did not clearly explain the movement deviation regarding the influence of gravity. In the revised version, we have re-formulated Equation (1) (page 6, line 156).

Edit 1: “This can be explained by the force analysis^{11,24}:

$$\begin{aligned} f_{\text{kinetic}} &= \hat{\mu}_{\text{wat_suc}} F_{\text{contact}}, \\ F_{\text{contact}} &= F_{\text{bend}}, \\ F_{\text{bend}} &= (p_{\text{atm}} - p_{\text{in}})A - F_{\text{pull}} + mg, \end{aligned} \tag{1}$$

where m is the mass of the suction cup. Here we assume the suction cup is place on a horizontal substrate, and will discuss the situation when the substrate is tilted later. ...”

Reply 2: Our claim “moments generated by the robot's weight are neglectable” is made in the context of analysing the force balance at the suction interface (page 8, line 217). Here, the “moment” we refer to is

the tilting moment, i.e., the mass of the robot times the distance of centre-of-mass from the substrate. The tilting moment applied to the suction cup will change the suction cup's contacting pressure distribution on the substrate. However, we assume the kinetic coefficient of friction of the hydrophilic silicone-water-substrate interface is constant, i.e., not affected by the contacting pressure change. Therefore, we can assume that the tilting moment does not affect the total contacting force of the suction cup applying to the substrate (i.e., F_{bend}). Moreover, since 1. The robot is lightweight (~96 g); 2. The distance between the centre of mass of the robot and the substrate is small ~20 mm (i.e., the robot is short in the direction perpendicular to the substrate), and 3. The suction cup diameter is relatively large (50 mm), the influence of the tilting moment is further reduced. We have made changes in the main text (page 8, line 217) for clarity.

Edit 2: “The influence of the tilting moment (i.e., the mass of the robot times the distance of centre-of-mass from the substrate) is neglectable. This is because it affects the contacting pressure distribution but not the total contacting force at the suction interface and we assume the CoF is constant and not affected by the pressure change. In addition, the robot is lightweight (96 g), has low centre-of-mass (~20 mm) and a relatively large suction cup (diameter of 50 mm), further reducing the influence of the tilting.”

Reply 3: Although we do not think weight-induced tilt is the cause of movement deviation, we suspect gravity and weight of the robot (not in the form of tilting moment) do cause the deviation. We separately discuss the deviation of rotation and translation movement cases as below.

For translation movement (i.e., upward climbing, downward and horizontal sliding), the weight of SSR affects its performance, according to Fig. 3F. According to our calculation, provided in **Supplementary Information S6.2 Climbing ability**, the weight of the robot adds more resisting torque to the motors when climbing upward, while inversely mitigates motors' load when sliding downward – note that we always apply the same voltage to drive the motors in open-loop control. This is demonstrated by the higher downward speed and lower upward speed in Fig. 3F. In addition, upward climbing exhibits larger deviation than downward sliding. We think this is because water cannot easily or rapidly wet (via capillary force) the front of the suction cup when acting against gravity, which may cause some dry friction at the suction interface.

For rotation movement, rather than the effect of gravity due to the SSR's mass, we believe the deviation is due to gravity affecting the movement and retention of water. From Fig. 3C we see that on tilted substrates the SSR has larger movement deviation than on horizontal substrates and there is no significant difference between the 45°, 90° and 135° cases, but these exhibit larger deviation than 0° and 180°. Therefore, we suspect that the gravity effect on water might be the main cause. On all the three tilted cases,

water can flow downward, which might wet one tyre and cause slip while the wheel opposite to the water flow remains dry. The unbalanced friction causes the movement deviation. If, however, the weight of the robot body were the main cause, the 90° case should show a larger deviation than the 45° and 135° cases. Therefore, we suspect that the light weight of SSR (~90 g if the portable water tank is not equipped) does not significantly affect the rotation movement.

Edit 3: “For rotation (Fig. 3C), the SSR achieves smaller centre deviation in balanced-gravity cases (0° and 180°) than unbalanced-gravity cases (45°, 90° and 135°). It was observed that some water flowed downward by the effect of gravity and wetted one or more of the tyres contacting the substrate. Imbalance of friction due to this wetting, and subsequent tyre slip, leads to movement deviation. The weight of the SSR (~90 g when the portable water tank is not equipped) does not show significant influence on the rotation movement deviation since no clear difference between 45°, 90° and 135° cases was observed. For translation (Fig. 3F), the SSR’s speed and deviation are, however, affected by gravity and its weight. According to the calculation in Supplementary Information S6, it is clear that the weight of SSR adds extra resisting torque for the motors to overcome when the SSR climbs upward (sliding speed 0° ≈ 180° > 45° ≈ 135° > 90°), but reduces the motors’ load when the SSR slides downward (sliding speed 90° > 45° ≈ 135° > 0° ≈ 180°). Gravity component acting against the sliding direction can preclude water from fully wetting the front edge of the moving robot, thereby increasing the likelihood of stuck movement as described above. In contrast, the gravity component acting in the sliding direction accelerates water to wet the head of the suction cup. This explains why the upward climbing has larger deviation than downward sliding.”

Major comment #2: The authors noted that "the suction cup adhesion does not require any energy." This seems not true. Any suction cup can't avoid the air leak. Is there no air leak in your suction cup? How do you prevent this air leak?

Reply 1: Thank you for this comment. Indeed, no suction cup can avoid leakage. Here, the application scenario of the proposed suction cup is on smooth surfaces, for example, glass and PMMA sheets. These surfaces are common in real world, for example, the outer glass wall of skyscrapers, the blades of wind turbine, the hull of ships, the tanks in chemical plants and the tiles of swimming pools, which endow the SSR with great potential for practical applications. In addition, we fabricated the suction cup by 3D printing the mould with ABS and smoothing the mould surface with acetone. The cast silicone pad has a smooth bottom surface; therefore, the leakage between the smooth substrate and smooth suction cup surface is slow and neglectable. As we tested, the robot can adhere to the PMMA sheet upside down overnight, which is long enough for the most practical requirements. This is also consistent with our

experiences – we did not see any suction failure during the experiments. Therefore, it is rational to regard the practical leakage rate as zero during the sliding. In this case, the energy for maintaining the suction is zero. For clarity, we have added following explanations in the main text (page 13, line 391).

Edit 1: ... “When the robot is statically adhering to the substrate, the power consumption is 0 W (all the electronics turned off). This is because energy is only required for driving the sliding movement, secreting water and attaching to/detaching from the substrate, and the suction adhesion requires zero energy on smooth surfaces since the leakage on smooth surfaces is neglectable. Based on our test, the SSR stably adheres upside-down to the PMMA surface overnight, which is long enough for practical applications.” ...

Reply 2: However, suction is more challenging on rough surfaces, and we believe this is the focus of your comment. To better understand the influence of surface roughness on sliding suction performance of SSR, we have performed new experiments and added new discussions in Supplementary Information S9 (page 7, line 186). We engraved PMMA sheets with thin grooves with different depths and let SSR slide on these surfaces. Suction pressure was measured during the sliding. Experimental results show that the surface roughness reduces the time that the SSR can maintain sliding suction and that sliding movement on rough surfaces increases the leakage rate by approximately a half. To mitigate this leakage loss, a small negative pressure pump can be used. We have theoretically demonstrated that the pump-assisted SSR consumes much lesser ($O(10^{-4})$) energy than traditional pump-assisted wall-climbing robots. The experimental results on rough surfaces and theoretical analysis have been provided in a new section **Supplementary Information S9 Discussion of the pump-assisted design for sliding on rough surfaces** (page 7, line 186) as below.

Edit 2:

“S9 Discussion of the pump-assisted design for sliding on rough surfaces

To determine the influence of surface roughness on sliding suction performance, we conducted a series of experiments on rough surfaces. Three rough surfaces were prepared by engraving flat PMMA sheets. We use a FusionEdge (EpilogLaser) laser cutter to engrave parallel lines (0.5 mm distance) with different depths. From surface 1 (denoted as S1, smoothest) to surface 3 (denoted as S3, roughest), we set the engraving laser parameters as 100 % speed, 5 / 10 / 15 % power and 5000 Hz. The engraved surfaces are shown in Supplementary Fig. 7A. To measure the pressure inside the suction region, we drilled a through-hole (1 mm diameter) on the bottom centre of the suction cup, and connected the hole with thin silicone tubes to a pressure sensor (SSCDRRV015PDAA5, Honeywell). Data was read by an Arduino Nano board. Experiments were conducted via following steps: First, we wet the rough surface with a thin film of water. Second, we manually squeezed the SSR to the surface then released it, from which the suction is deemed

to begin (as the black dashed line shows in Supplementary Fig. 7B). We separate the suction into two states: static and sliding. For static suction, we did not activate the movement of SSR; for the sliding suction, we activate the forward sliding movement parallel to the grooves at the moment of release (black dashed line). According to Supplementary Fig. 7B, surface roughness reduces both the maximum suction pressure and the suction longevity. In addition, the sliding movement does not affect the maximum suction pressure on rough surfaces compared with the static state; however, the suction longevity is reduced by approximately 50 %. During the sliding, new percolation channels between the environment and the suction region are introduced by the relative movement of the two contacting surfaces. We attribute the longevity reduction caused by the sliding movement to this dynamic change in leakage channels. In contrast, the percolation channels of a static suction cup are relatively stable and constant, and their number and form only change by leakage-induced shape recovery of the suction cup. We will not analyse more on the micro contact mechanics⁴⁻⁶ since this is not the focus of this paper; however, given the experimental results in Supplementary Fig. 7, we can draw the conclusion that the sliding-induced leakage is in the same order of magnitude as the roughness-induced leakage.

Supplementary Fig. 7. Sliding suction test on rough surfaces. (A) Rough grooved surfaces prepared by laser engraving. Scale bar: 1 mm. (B) Measured suction pressure on rough surfaces in static and sliding suction situations.

However, this does not mean that the sliding suction cannot be applied to rough surfaces. Additional mitigation can be employed to maintain suction; for example, a small vacuum pump can be connected to the suction region to compensate for leakage. This design is similar to traditional wall-climbing robots^{7,8}, however, the SSR will consume much less energy according to our analysis below. Previous pump-assisted wall-climbing robots require the body to be lifted 1-5 mm (typically 5 mm for commercial products⁹) from the substrate to avoid direct contact and friction; however, this causes severe leakage and accordingly needs a strong vacuuming flow. In contrast, the sliding suction method employs direct body contact with

the substrate. The direct contact with a rough surface (e.g., coarse sandpaper with grit >40) leaves a mean gap of $\sim 200 \mu\text{m}$ (we use half of the granule size of the 40-grit sandpaper for estimation)¹⁰. According to the leakage model¹¹, the leakage rate Q of a leaking suction cup can be estimated by

$$Q = \frac{L_y d^3 (p_a - p_b)}{12\eta L_x},$$

where L_x and L_y are the width and length of the sealed region of the suction cup, p_a and p_b are the pressure of the outer environment and the internal suction region, d is the gap width (for the pump-assisted wall-climbing robot, we use $d = 5 \text{ mm}$; for the pump-assisted sliding suction robot, we use $d = 200 \mu\text{m}$), η is the viscosity of the fluid (for the pump-assisted wall-climbing robot, the fluid is air so $\eta = 5 \times 10^{-5} \text{ Pa}\cdot\text{s}$; for the pump-assisted sliding suction robot, the fluid is water so $\eta = 1 \times 10^{-3} \text{ Pa}\cdot\text{s}$). Assuming they both require the same pressure differential to adhere to the substrate (typically $p_a - p_b = 2.5\sim 4.5 \text{ kPa}$ ^{7,8}) and have the same size (i.e., same L_y/L_x), a simple calculation can derive the ratio of the flow rate:

$$\frac{Q_{\text{centrifugal}}}{Q_{\text{sliding_suction}}} = \frac{d_{\text{centrifugal}}^3 \eta_{\text{water}}}{d_{\text{sliding_suction}}^3 \eta_{\text{air}}} = 3.125 \times 10^5$$

The ratio of the power consumption for maintaining the suction is:

$$\frac{P_{\text{centrifugal}}}{P_{\text{sliding_suction}}} = \frac{Q_{\text{centrifugal}} d_{\text{centrifugal}} \rho_{\text{air}}}{Q_{\text{sliding_suction}} d_{\text{sliding_suction}} \rho_{\text{water}}} \approx 1.01 \times 10^4.$$

This indicates that the power consumption on a rough surface (~ 40 grit) of a pump-assisted sliding suction robot is four orders of magnitude less than a regular pump-assisted wall-climbing robot, demonstrating a significant advance. In addition, we also consider comparing the pump-assisted sliding suction robot with wall-climbing robots based on the Bernoulli effect, which uses a reverse high-speed airflow. A Bernoulli-effect-based climbing robot requires a much higher flow rate to generate the pressure drop than suction-based robots, which consumes significantly more energy¹².”

Reply 3: In summary, our answer to your question “**Is there no air leak in your suction cup? How do you prevent this air leak?**” is that the goal of the current design is to apply the SSR on smooth surfaces, on which the leakage can be neglected; for future applications on rough surfaces, the SSR can use a small additional pump to help maintain the suction pressure. In the main text, we have added following text in **Discussions** (page 14, line 400).

Edit 3: “The current design of SSR can only slide on relatively smooth surfaces. Surface roughness will reduce the sliding suction performance since it will cause leakage and generate higher friction. However, it will increase the static friction on tyres to avoid slippage. Future designs can use a small additional vacuuming pump to compensate for the leakage during sliding on rough surfaces. A pump-assisted SSR

will be much more energy efficient than regular pump-assisted wall-climbing robots (and is predicted to use approximately four orders of magnitude less energy), thanks to the smaller leaking orifice. Detailed discussions are provided in Supplementary Information S9.” ...

Major comment #3: The known facts may not be necessary for the manuscript, such as the explanation of the game controller in lines 272-277 (it does not affect the contents of the paper), and subsection 4.1 "movement deviation." You can use the space to explain the unclear part, like the equation (6) (how do you have it?) , the experimental results shown in figure 5 (why is the movement deviation for the rotation motion in 90 Deg case smaller than 45 Deg and 135 Deg? Why does the robot move sideway even driving it up and down in 0 Deg and 180 Deg?)

Reply 1: Thank you for this suggestion. We have removed the descriptions of the game controller into **Supplementary Information S8 Control the SSR via a wireless game controller** (page 7, line 178). We have also removed some unnecessary descriptions of movement deviation. According to your suggestions, we have added additional text to explain Equation (6) (now Eq. (5) in the revised version, page 9, line 248).

Edit 1: “According to $f_{\text{kinetic}} = \hat{\mu}_{\text{wat_suc}} F_{\text{bend}}$ in Eq. (1), once we got $\hat{\mu}_{\text{wat_suc}}$ and the $f_{\text{kinetic}} - F_{\text{pull}}$ relation (Fig. 1F), the $F_{\text{bend}} - F_{\text{pull}}$ relation can be calculated as

$$F_{\text{bend}} = 0.12F_{\text{pull}} + 1.54. \quad (5)$$

Eq. (5) indicates that the increase of F_{pull} slightly increases F_{bend} ; therefore, f_{kinetic} is further slightly influenced by F_{pull} . The decrease of the internal pressure p_{in} , which does not consume energy, balances most of the increase of F_{pull} , according to $F_{\text{pull}} = (p_{\text{atm}} - p_{\text{in}})A - F_{\text{bend}}$ in Eq. (1). Although the decrease of p_{in} makes the suction cup squeeze harder, it does not change the geometry very much; this is why F_{bend} only increases slightly and the sliding suction mechanism has such high loading ability. Eq. (1), (2), (3) and (5) are the critical formulae of the physical model of the SSR, through which we calculated the sliding ability, climbing ability and loading ability of the SSR. Details of the formula derivation are provided in Supplementary Information S6. Calculation results suggest that the SSR can easily slide upside-down, climb on the wall and carry a heavy mass.”

Reply 2: For your question “**why is the movement deviation for the rotation motion in 90 Deg case smaller than 45 Deg and 135 Deg?**”, we have repeated the experiment for three times in the revised manuscript. The results show that the 90-degree, 45-degree and 135-degree cases do not show significant difference regarding the deviation of rotation movement, but they are all larger than the 0-degree and 180-degree cases. We think our **Reply to your Major Comment #1** can explain this phenomenon. For your question “**Why does the robot move sideway even driving it up and down in 0 Deg and 180 Deg?**”, We apologise

for the confusion and agree that Fig. 3E was not clear, leading to your observation. We have re-drawn Fig. 3E to reduce ambiguity and show that there is almost zero sideways movement for the 0 and 180 Degree cases. We have also explained this more clearly in the text and figure caption.

Edit 2: “For 0° and 180° cases, we actuated the robot to slide horizontally and draw the horizontal trajectories on the ordinate axis in Fig. 3E for ease of comparison with other cases.”

Fig. 3. Experimental characterization of the sliding suction ability of the SSR. ... (E) Translational trajectories of the robot on substrate with five different tilted angles. For the horizontal trajectories of 0° and 180° cases, we align movement with the ordinate axis. ...

Major comment #4: The accuracy of the user-controlled route tracking must depend on the experience

and skill of the driver. It is not clear what performances need to be evaluated.

Reply: Thank you for this comment. The control of the robot is very smooth and easy to use, just like a remote-control car toy. It is not easy to quantitatively evaluate a driver's skill; therefore, we can only qualitatively record it. To prove the SSR is easy-to-use, we have implemented a series of additional experiments (page 11, line 318). Three volunteers with no knowledge about the sliding suction mechanism were recruited. They were informed of the basic functions of the controller. They were asked to use the PS4 game controller to drive the SSR (with 200 g payload) mounted upside-down to the lower side of the PMMA sheet, from the starting point to the destination, avoiding obstacles. All three volunteers successfully completed the experiment. Therefore, we believe it demonstrates that the proposed SSR is easy to use, and has potential for delivering objects in industry. In addition, we performed another experiment for using SSR as a cleaning robot to further demonstrate its practical applications.

Edit: “Demonstration: user-controlled weight transportation with obstacle avoidance. We recruited three volunteers without knowledge of sliding suction. Volunteers were taught the functions of the buttons on the game controller, and were required to use the game controller to drive the SSR (loaded with 200 g mass) and avoid obstacles from the starting point to the destination. The diagram of experimental setup is shown in Fig. 4B i. All volunteers successfully and smoothly completed the task, as shown in Fig. 4B ii-iv and Supplementary Movie 4.

...

Demonstration: erasing marker on different surfaces. A mop (fabricated by attaching a sponge to a 3D-printed bracket, as shown in the insert of Fig. 4D i) was attached to the SSR. The mop was soaked with isopropanol. Two surfaces, an upright painted metal wall and a PMMA ceiling, were drawn with a cross using a permanent marker pen. The SSR was remotely controlled to carry the mop to approach the permanent marker and then erase it using the mop. The SSR successfully completed the task on both the upright wall and ceiling, as shown in Fig. 4D and Supplementary Movie 4.”

Fig. 4. Demonstration of applications of SSR. ... (B) User-controlled upside-down sliding with 200 g payload and obstacle avoidance. i: diagram of the experimental setup. ii to iv: three volunteers successfully controlled the SSR to slide from the starting point to the destination. ... (D) Demonstration of using the SSR as a carrier to clean markers on an upright painted metal wall (i) and an upside-down PMMA ceiling (ii).

Major comment #5: For the S5 "Measurement of the suction cup bending force. ...", it is valuable to refer paper (Dingxin Ge, Takahiro Matsuno, Yi Sun, Chao Ren, Yongchen Tang, and Shugen Ma, Quantitative Study on the Attachment and Detachment of a Passive Suction Cup, Vacuum, 116, 2015, pp.13/20)

Reply 1: Thank you for this comment. We have carefully read this paper and have added the citation in **main text** (page 6, line 156).

Edit 1: ... "This can be explained by the force analysis^{11,24};" ...

Reply 2: And cited in **Supplementary Information** (page 6, line 147).

Edit 2: "The force characterization method of suction cups has been well reported^{11,24};" ...

Minor comment #1: It is understandable that "SSR untethered (Fig .6A)" in line 228

Reply: Thank you for this comment. We wanted to say that "In its further design, the SSR can be untethered. i.e., equipped with an on-robot water tank to supply water, as shown in ...". We have rephrased the description for clarity (page 7, line 190).

Edit: ... "A portable water tank (not shown in Fig. 3A) can be mounted onto the robot (Supplementary Fig. 8A); then the SSR can be completely untethered." ...

Minor comment #2: You seem to use $g = 10$ for the explanation in line 190. For a scientific paper, $g = 9.8$, at least, should be used.

Reply: We appreciate this observation. We have used $g = 9.8$ in the revised manuscript and recalculated the equations. It did not affect the conclusion results.

Minor comment #3: The robot's weight is not considered in equation (1), but there is no explanation.

Reply: Thank you for this comment. We have included the suction cup's weight in Equation (1) and re-derived the formulae (page 6, line 157).

Minor comment #4: For S6.3 "the Loading ability" means the maximum suction force of the cup, but not pulling the payload up. Please check this.

Reply: Thank you for this comment. We agree that the "loading ability" is not a suitable description. In the revised manuscript, we have changed all "loading ability" to "payload ability".

Reviewer #3,

General comment: Report on "Water-enhanced..." by T. Yue et al., I think this is an interesting study but in its present form of limited practical usefulness.

Reply: The authors appreciate you very much for reviewing the manuscript and providing valuable comments to help us significantly improve the quality. Based on your comments, we have made following major changes:

1. We have added two more experiments to demonstrate the practical applications.
2. We have implemented additional sliding suction experiments on rough surfaces, and discussed more on the difficulties and potential of achieving sliding suction on rough surfaces in future.
3. We have made other changes according to your detailed comments below.

In particular, we have aimed to more clearly explain our thinking of the practical usefulness of the proposed sliding suction mechanism and its robotic applications. Although the current form has been most demonstrated on smooth surfaces (as you questioned in **Comment #1**), we believe it will not preclude the sliding suction from practical use. In real world there are many surfaces, especially artificial surfaces, are smooth. For example, the outer glass wall of skyscrapers, the blades of wind turbines, the hull of ships, the tanks in chemical plants, the tiles of swimming pools, etc. These surfaces need cleaning, maintenance and inspection; however, these surfaces are difficult to access and challenging for state-of-the-art robotic climbing technologies. Even just the window cleaning robot market is expected to reach \$264.18 million by 2030, growing at a CAGR of 15.2%²⁵. Based on our knowledge, no other method can realise an untethered robot which can climb on these surfaces with low cost, easy fabrication, low energy consumption and high payload capacity. We believe the proposed sliding suction mechanism has great potential for these practical applications.

In addition, equipping an additional small vacuum pump to compensate for the leakage can enable the sliding suction robot slide on rough surfaces in future. We will discuss this in detail in the **Reply to Comment #1**.

Therefore, we believe the sliding suction method will have many practical applications. In the revised manuscript, we have also implemented two new demonstrations, together with previous two experiments, to provide more practical applications (page 11, line 318).

Edit: "Demonstration: user-controlled weight transportation with obstacle avoidance. We recruited three volunteers without knowledge of sliding suction. Volunteers were taught the functions of the buttons on the game controller, and were required to use the game controller to drive the SSR (loaded with 200 g mass) and avoid obstacles from the starting point to the destination. The diagram of experimental setup is

shown in Fig. 4B i. All volunteers successfully and smoothly completed the task, as shown in Fig. 4B ii-iv and Supplementary Movie 4.

...

Demonstration: erasing marker on different surfaces. A mop (fabricated by attaching a sponge to a 3D-printed bracket, as shown in the insert of Fig. 4D i) was attached to the SSR. The mop was soaked with isopropanol. Two surfaces, an upright painted metal wall and a PMMA ceiling, were drawn with a cross using a permanent marker pen. The SSR was remotely controlled to carry the mop to approach the permanent marker and then erase it using the mop. The SSR successfully completed the task on both the upright wall and ceiling, as shown in Fig. 4D and Supplementary Movie 4.”

Fig. 4. Demonstration of applications of SSR. ... (B) User-controlled upside-down sliding with 200 g payload and obstacle avoidance. i: diagram of the experimental setup. ii to iv: three volunteers successfully controlled the SSR to slide from the starting point to the destination. ... (D) Demonstration of using the SSR as a carrier to clean markers on an upright painted metal wall (i) and an upside-down PMMA ceiling (ii).

Comment #1: The proposed set-up will only work on relative smooth surfaces as otherwise the suction cup will not seal.

Reply 1: Thank you for this comment. We agree that the present form of SSR works most effectively on relative smooth surfaces. Surface roughness will affect the sliding suction performance from two aspects: for suction, roughness will cause leakage, which causes suction failure; for sliding, a rough surface might generate higher friction force, reducing the sliding movement in comparison with a smooth surface. However, roughness is also expected to increase the static friction on wheel tyres and thereby reduce slip. In general, rough surfaces are more challenging for the SSR.

To better understand the influence of surface roughness on sliding suction performance of SSR, we have performed new experiments and added new discussions. We engraved PMMA sheets with thin grooves with different depth and let SSR slide on these surfaces. Suction pressure was measured during the sliding. Experimental results show that the surface roughness reduces the operating time of the SSR; in addition, the sliding movement also increases the leakage rate by approximately half. However, leakage can be reduced by adding a small negative pressure pump. We have theoretically demonstrated that the pump-assisted SSR consumes much lesser ($O(10^{-4})$) energy than traditional pump-assisted wall-climbing robots. The experimental results on rough surfaces and theoretical analysis have been provided in a new section **Supplementary Information S9 Discussion of the pump-assisted design for sliding on rough surfaces** (page 7, line 186) as below.

Edit 1: “S9 Discussion of the pump-assisted design for sliding on rough surfaces

To determine the influence of surface roughness on sliding suction performance, we conducted a series of experiments on rough surfaces. Three rough surfaces were prepared by engraving flat PMMA sheets. We use a FusionEdge (EpilogLaser) laser cutter to engrave parallel lines (0.5 mm distance) with different depths. From surface 1 (denoted as S1, smoothest) to surface 3 (denoted as S3, roughest), we set the engraving laser parameters as 100 % speed, 5 / 10 / 15 % power and 5000 Hz. The engraved surfaces are shown in Supplementary Fig. 7A. To measure the pressure inside the suction region, we drilled a thorough-hole (1 mm diameter) on the bottom centre of the suction cup, and connected the hole with thin silicone tubes to a pressure sensor (SSCDRRV015PDAA5, Honeywell). Data was read by an Arduino Nano board. Experiments were conducted via following steps: First, we wet the rough surface with a thin film of water. Second, we manually squeezed the SSR to the surface then released it, from which the suction is deemed to begin (as the black dashed line shows in Supplementary Fig. 7B). We separate the suction into two states: static and sliding. For static suction, we did not activate the movement of SSR; for the sliding suction, we activate the forward sliding movement parallel to the grooves at the moment of release (black

dashed line). According to Supplementary Fig. 7B, surface roughness reduces both the maximum suction pressure and the suction longevity. In addition, the sliding movement does not affect the maximum suction pressure on rough surfaces compared with the static state; however, the suction longevity is reduced by approximately 50 %. During the sliding, new percolation channels between the environment and the suction region are introduced by the relative movement of the two contacting surfaces. We attribute the longevity reduction caused by the sliding movement to this dynamic change in leakage channels. In contrast, the percolation channels of a static suction cup are relatively stable and constant, and their number and form only change by leakage-induced shape recovery of the suction cup. We will not analyse more on the micro contact mechanics⁴⁻⁶ since this is not the focus of this paper; however, given the experimental results in Supplementary Fig. 7, we can draw the conclusion that the sliding-induced leakage is in the same order of magnitude as the roughness-induced leakage.

Supplementary Fig. 7. Sliding suction test on rough surfaces. (A) Rough grooved surfaces prepared by laser engraving. Scale bar: 1 mm. (B) Measured suction pressure on rough surfaces in static and sliding suction situations.

However, this does not mean that the sliding suction cannot be applied to rough surfaces. Additional mitigation can be employed to maintain suction; for example, a small vacuum pump can be connected to the suction region to compensate for leakage. This design is similar to traditional wall-climbing robots^{7,8}, however, the SSR will consume much less energy according to our analysis below. Previous pump-assisted wall-climbing robots require the body to be lifted 1-5 mm (typically 5 mm for commercial products⁹) from the substrate to avoid direct contact and friction; however, this causes severe leakage and accordingly needs a strong vacuuming flow. In contrast, the sliding suction method employs direct body contact with the substrate. The direct contact with a rough surface (e.g., coarse sandpaper with grit >40) leaves a mean gap of $\sim 200 \mu\text{m}$ (we use half of the granule size of the 40-grit sandpaper for estimation)¹⁰. According to the leakage model¹¹, the leakage rate Q of a leaking suction cup can be estimated by

$$Q = \frac{L_y d^3 (p_a - p_b)}{12\eta L_x},$$

where L_x and L_y are the width and length of the sealed region of the suction cup, p_a and p_b are the pressure of the outer environment and the internal suction region, d is the gap width (for the pump-assisted wall-climbing robot, we use $d = 5$ mm; for the pump-assisted sliding suction robot, we use $d = 200$ μm), η is the viscosity of the fluid (for the pump-assisted wall-climbing robot, the fluid is air so $\eta = 5 \times 10^{-5}$ Pa·s; for the pump-assisted sliding suction robot, the fluid is water so $\eta = 1 \times 10^{-3}$ Pa·s). Assuming they both require the same pressure differential to adhere to the substrate (typically $p_a - p_b = 2.5 \sim 4.5$ kPa^{7,8}) and have the same size (i.e., same L_y/L_x), a simple calculation can derive the ratio of the flow rate:

$$\frac{Q_{\text{centrifugal}}}{Q_{\text{sliding_suction}}} = \frac{d_{\text{centrifugal}}^3 \eta_{\text{water}}}{d_{\text{sliding_suction}}^3 \eta_{\text{air}}} = 3.125 \times 10^5$$

The ratio of the power consumption for maintaining the suction is:

$$\frac{P_{\text{centrifugal}}}{P_{\text{sliding_suction}}} = \frac{Q_{\text{centrifugal}} d_{\text{centrifugal}} \rho_{\text{air}}}{Q_{\text{sliding_suction}} d_{\text{sliding_suction}} \rho_{\text{water}}} \approx 1.01 \times 10^4.$$

This indicates that the power consumption on a rough surface (~40 grit) of a pump-assisted sliding suction robot is four orders of magnitude less than a regular pump-assisted wall-climbing robot, demonstrating a significant advance. In addition, we also consider comparing the pump-assisted sliding suction robot with wall-climbing robots based on the Bernoulli effect, which uses a reverse high-speed airflow. A Bernoulli-effect-based climbing robot requires a much higher flow rate to generate the pressure drop than suction-based robots, which consumes significantly more energy¹².”

Reply 2: In the **main text** of the revised version, we have added following explanations in the **Discussion** (page 14, line 400).

Edit 2: “The current design of SSR can only slide on relatively smooth surfaces. Surface roughness will reduce the sliding suction performance since it will cause leakage and generate higher friction. However, it will increase the static friction on tyres to avoid slippage. Future designs can use a small additional vacuuming pump to compensate for the leakage during sliding on rough surfaces. A pump-assisted SSR will be much more energy efficient than regular pump-assisted wall-climbing robots (and is predicted to use approximately four orders of magnitude less energy), thanks to the smaller leaking orifice. Detailed discussions are provided in Supplementary Information S9.” ...

Comment #2: To move the “robot” use wheels but the snail can move just using body movements. Would it be possible to change the design to operate more like the snail without external wheels. It would be interesting to know how the snail is able to move.

Reply 1: Thank you for this comment. The snail's movement is interesting – it can dexterously control the abdominal muscles to generate strong and stable suction for adhesion²⁶ and peristalsis for movement²⁷. The abdominal muscles can simultaneously contract and relax at multiple sites. The contraction forms pedal waves, by which the whole abdomen can be divided into parallel distributed segments. The travelling waves move from the tail to the head, by which the snail can slide forward. The snail's mucus is a non-Newtonian fluid²⁷, which becomes a lubricant in high pressure and solid in low pressure. The musculature contraction and relaxation apply high and low pressure to the mucus, helping the snail utilise friction at the rear and lubrication at the front of the travelling wave to move forward. All these biological structures and behaviours make the snail slide via its body movements. We agree that the sliding suction movement of snails may be more effective than our proposed artificial sliding suction strategy, since it exhibits a more compact size, higher power density and more dexterous control.

However, we cannot design a climbing robot with the same movement principle as a snail since the dexterous musculature movement requires a soft artificial muscle with high degrees of freedom and a complex control algorithm, such as a stacked dielectric elastomer actuator (DEA). In this manuscript, we focus on introducing the concept of sliding suction; therefore, we use motors and wheels – an easy-to-access method – to generate the movement. We also hope we can combine the snail's soft body movement with the sliding suction mechanism, which is the next goal of our work. In the manuscript, we have explained more about the snail's body movement in the **Principle of sliding suction mechanism**. (page 3, line 74).

Edit 1: “Snails secrete mucus to lubricate the contacting interface and reduce the coefficient of friction (CoF), making sliding movement possible. In addition, mucus can enhance the suction by sealing the gaps between the abdomen and the substrate³. Coordinated body movements, in the form of travelling waves, then generate the driving force to slide forward. Other low-viscous liquids, such as water, also reduce friction, which previous studies in contact and friction mechanics have well reported¹.”

Reply 2: And modified the diagram of Fig. 1A to make it clear.

“Fig. 1. The sliding suction mechanism. (A) The snail achieves sliding suction by mucus secretion and muscles’ movement. IW: interwave. PW: pedal wave.” ...

Comment #3: The authors mention that Gecko adhesives are problematic on contaminated surfaces. But I believe this is the case also for the suction cup mechanism suggested in this study. If the substrate contains e.g. dust and sand particles then the suction cup is unlikely to seal the contact.

Reply: Thank you for this comment. We agree that if the surface is contaminated, the robot might also not be able to form a seal. We have removed related text in the revised version.

Comment #4: Another problem is if there is a step-like change in the surface topography. The sliding suction cup cannot move over a step-like change in the surface height even in the downhill direction because it would not seal when overlapping the step. On the other hand a robot with several suction cups moving step-like like humans could move over a step.

Reply: Thank you for this suggestion. We agree that the step-like change in surface topography is a big challenge for the sliding suction. A possible solution is, as you mentioned, using a multi-suction-cup system to move over a step. We have added this as one of the limitations of the sliding suction in **Discussion** (page 14, line 406); however, we cannot propose a multi-suction-cup system in this work due to limited time and space. It is an interesting topic, and we hope we can achieve this goal in our future

work.

Edit: “In addition, the SSR cannot slide over a step-like change on the surface. Future design can use multiple suction cups to move such discrete topographic features.”

Comment #5: The physics of suction cups has been studied in Ref. “Physics of suction cups”, A Tiwari, BNJ Persson, *Soft matter* 15 (46), 9482-9499 (2019). The author may want to study this reference. The same authors have also published an article about suction cups in water.

Reply 1: Thank you for recommending this paper, and this is the paper we have studied for a long time. We are happy to cite it in this manuscript (page 6, line 156).

Edit 1: ... “This can be explained by the force analysis^{11,24}.” ...

Reply 2: and in **Supplementary Information** (page 6, line 147).

Edit 2: “The force characterization method of suction cups has been well reported^{11,24}.” ...

Comment #6: The rubber wheels are made from silicone rubber and the authors state that this result in large friction also in water due to dewetting. But this is not necessarily the case if the substrate is wetted by water as in that case the interface will be hydrophilic even if the wheel surface is hydrophobic.

Reply 1: Thank you for this comment. We would like to explain it as follows. Before the wheel contacts with water, air was in the small asperities of the rubber wheel. These bubbles will be trapped on the wheel surface when the rubber wheel is instantaneously submerged in water, and be sandwiched in the middle of the suction interface. This is like air trapped in the cup when we submerge a cup underwater with the opening downward. The sandwiched air act to maintain small regions of contact in the dry condition with high friction force. Therefore, the wheel can maintain relatively high friction with the substrate even if the substrate is pre-wetted by water. This feature makes the robot able to pass over its previous water tracks. We have added following explanations in **Designing water-enhanced sliding suction mechanism** (page 4, line 97).

Edit 1: “In contrast, the plain silicone pad remains naturally hydrophobic and repels water from the interface, even if the hydrophobic silicone is placed on a pre-wetted surface. This is because the instantaneous submergence in water cannot let the tiny air bubbles trapped in the silicone surface disappear; therefore, these bubbles still remain between the silicone and the substrate, making the interface more like a dry contact and generating a high friction force (Fig. 1B i to iii).”

Reply 2: However, we also find that when the wheel contacts with water too much times, tyre slip sometimes occurs. This might be because air is gradually squeezed out by the constant contact between the wheel and substrate, and water gradually gets into the rubber wheel surface asperities. This explains why the robot has movement deviation in the rotation test and we found the robot sometimes cannot move

smoothly. To avoid the slippage on wetted surface, a possible solution could be coating the rubber wheel with hydrophobic spray to make the surface more hydrophobic. We have added additional descriptions in **Discussion** (page 12, line 347).

Edit 2: “We noticed that when the wheel contacts with water too much, the wheel’s friction reduces. This might be because water slowly fills the rubber wheel surface asperities, and the air is squeezed out, making the hydrophobicity of wheels lose efficacy.”

Reference

- 1 Pawlak, Z., Urbaniak, W. & Oloyede, A. The relationship between friction and wettability in aqueous environment. *Wear* **271**, 1745-1749 (2011).
- 2 Wang, Y. *et al.* Water as a “glue”: Elasticity-enhanced wet attachment of biomimetic microcup structures. *Science Advances* **8**, eabm9341 (2022). <https://doi.org/doi:10.1126/sciadv.abm9341>
- 3 Smith, A. M. The biochemistry and mechanics of gastropod adhesive gels. *Biological adhesives*, 177-192 (2016).
- 4 Bottiglione, F., Carbone, G. & Mantriota, G. Fluid leakage in seals: An approach based on percolation theory. *Tribology International* **42**, 731-737 (2009).
- 5 Lorenz, B. & Persson, B. Leak rate of seals: Comparison of theory with experiment. *Europhysics Letters* **86**, 44006 (2009).
- 6 Persson, B., Albohr, O., Creton, C. & Peveri, V. Contact area between a viscoelastic solid and a hard, randomly rough, substrate. *The Journal of chemical physics* **120**, 8779-8793 (2004).
- 7 Ecovacs. *Ecovacs Window Cleaning Robot*, <<https://www.ecovacs.com/us/winbot-window-cleaning-robot/winbot-w1-pro>> (2023).
- 8 Fang, Y. *et al.* Design and optimization of wall-climbing robot impeller by genetic algorithm based on computational fluid dynamics and kriging model. *Scientific Reports* **12**, 9571 (2022). <https://doi.org/10.1038/s41598-022-13784-z>
- 9 Ecovacs. *Ecovacs Window Cleaning Robot Manual*, <https://static.ecovacs.com/upload/uk/file/support/2023/11/07/031356_9841-WINBOTWIPROManual.pdf> (2023).
- 10 Staff, G. E. *Sandpaper Grit Charts & Grades*, <<https://www.grainger.com/know-how/equipment/kh-video-sandpaper-grit-chart>> (2023).
- 11 Tiwari, A. & Persson, B. N. J. Physics of suction cups. *Soft Matter* **15**, 9482-9499 (2019). <https://doi.org/10.1039/C9SM01679A>
- 12 Wagner, M., Chen, X., Nayyerloo, M., Wang, W. & Chase, J. G. in *2008 IEEE/ASME International Conference on Mechatronic and Embedded Systems and Applications*. 210-215.
- 13 Shintake, J., Cacucciolo, V., Floreano, D. & Shea, H. Soft Robotic Grippers. *Advanced Materials* **30**, 1707035 (2018). <https://doi.org/10.1002/adma.201707035>
- 14 Yue, T. *et al.* A contact-triggered adaptive soft suction cup. *IEEE Robotics and Automation Letters* **7**, 3600-3607 (2022).
- 15 Gao, X., Yan, L., Wang, G. & Chen, I.-M. Modeling and analysis of magnetic adhesion module for wall-climbing robot. *IEEE Transactions on Instrumentation and Measurement* **72**, 1-9 (2022).
- 16 Hu, J., Han, X., Tao, Y. & Feng, S. A magnetic crawler wall-climbing robot with capacity of high payload on the convex surface. *Robotics and Autonomous Systems* **148**, 103907 (2022).
- 17 Li, S. *et al.* Switchable adhesion for nonflat surfaces mimicking geckos' adhesive structures and toe muscles. *ACS applied materials & interfaces* **12**, 39745-39755 (2020).
- 18 Guo, J., Leng, J. & Rossiter, J. Electroadhesion technologies for robotics: A comprehensive review. *IEEE Transactions on Robotics* **36**, 313-327 (2019).
- 19 Gu, G., Zou, J., Zhao, R., Zhao, X. & Zhu, X. Soft wall-climbing robots. *Science Robotics* **3**, eaat2874 (2018).
- 20 Menon, C., Murphy, M. & Sitti, M. in *2004 IEEE international conference on robotics and biomimetics*. 431-436 (IEEE).
- 21 Yoshida, Y. & Ma, S. in *2010 IEEE International Conference on Robotics and Biomimetics*. 1513-1518 (IEEE).
- 22 Hong, S., Um, Y., Park, J. & Park, H.-W. Agile and versatile climbing on ferromagnetic surfaces with a quadrupedal robot. *Science Robotics* **7**, eadd1017 (2022).
- 23 Yue, T., Bloomfield-Gadêlha, H. & Rossiter, J. Shape-Conformable Suction Cups with Controllable Adaptive Suction on Complex Surfaces. *IEEE Robotics and Automation Letters* (2023).
- 24 Ge, D. *et al.* Quantitative study on the attachment and detachment of a passive suction cup. *Vacuum* **116**, 13-20 (2015).

- 25 Ltd, C. D. I. P. *Window Cleaning Robot Market*, <[https://www.globenewswire.com/news-release/2023/05/11/2667126/0/en/Window-Cleaning-Robot-Market-Is-Expected-to-Reach-264-18-million-by-2030-Grow-at-a-CAGR-Of-15-2-during-Forecast-Period-2023-To-2030-Data-By-Contrive-Datum-Insights-Pvt-Ltd.html#:~:text=Farmington%2C%20May%2011%2C%202023%20\(,15.2%25%20From%202023%20To%202030.>](https://www.globenewswire.com/news-release/2023/05/11/2667126/0/en/Window-Cleaning-Robot-Market-Is-Expected-to-Reach-264-18-million-by-2030-Grow-at-a-CAGR-Of-15-2-during-Forecast-Period-2023-To-2030-Data-By-Contrive-Datum-Insights-Pvt-Ltd.html#:~:text=Farmington%2C%20May%2011%2C%202023%20(,15.2%25%20From%202023%20To%202030.>))> (2023).
- 26 Li, J., Peng, X., Ma, C., Song, Z. & Liu, J. Response mechanisms of snails to the pulling force and its potential application in vacuum suction. *Journal of the Mechanical Behavior of Biomedical Materials* **124**, 104840 (2021).
- 27 Pembury Smith, M. & Ruxton, G. D. How fast is a snail's pace? The influences of size and substrate on gastropod speed of locomotion. *Journal of Zoology* **314**, 12-19 (2021).

REVIEWER COMMENTS

Reviewer #1 (Remarks to the Author):

After reading the revised manuscript and the revision letter, I think that the manuscript has been improved in several aspects and most of my comments have been well-addressed. However, I still have some concerns that need to be further discussed before recommendation for acceptance. The specific comments are listed below:

1) During the tangential force test in Fig. 1D, the sliding suction is submerged in water tank. However, the practical condition of the sliding suction Fig. 4 is different. It is better to characterize the sliding suction based on the practical condition.

2) Both Fig. 1A and Fig. 2A shows the structure and function of the sliding suction of snail. However, from my side, the Fig. 2A may not be necessary because the locomotion of the robot is obvious different from the snail.

3) In Fig. 1E, the scale label of time is missing.

Reviewer #2 (Remarks to the Author):

Thank you for your hard work to address my concern. At this stage, I have no more questions.

Response Letter to Reviewer Comments

Dear Reviewers,

The authors would like to thank the reviewers for their efforts in assessing our manuscript. We have carefully checked all the reviewers' comments and made corresponding changes in the manuscript. In this response letter, **the reviewers' original comments are marked as red**, **our replies are marked as blue**, and **the changes made in the manuscript are in quotes and marked as purple**. Below we answer the reviewers' comments in detail.

Reviewer #1,

Reviewer's response: After reading the revised manuscript and the revision letter, I think that the manuscript has been improved in several aspects and most of my comments have been well-addressed. However, I still have some concerns that need to be further discussed before recommendation for acceptance. The specific comments are listed below:

Reply: We thank you for reviewing our manuscript again and believe your suggestions have helped us significantly enhance our work. Please see our detailed reply below.

Specific comment 1: During the tangential force test in Fig. 1D, the sliding suction is submerged in water tank. However, the practical condition of the sliding suction Fig. 4 is different. It is better to characterize the sliding suction based on the practical condition.

Reply: Thank you for this comment. We agree that practical sliding suction might be slightly different from the submerged-in-water condition, although our later sliding suction robot experiments show good performance. To be rigorous, we have implemented new characterizing experiments of the practical sliding suction condition as you suggest (i.e., pulling the SSR to slide on a dry PMMA surface and water being secreted by the robot). The results show that the measured sliding friction force of the SSR (i.e., the practical condition) is very close to the friction force of a hydrophilic suction cup submerged in the water tank. We have added the following text in the manuscript:

Edit: (page 6, line 145) "Note that Fig. 1E also shows the tangential force for the local wetting case – for the sliding robot – which will be discussed later."

Edit: (page 7, line 184) "... This water secreting system allows the robot to benefit from a local water environment (with water supply rate of approximately 0.05 mL/s when the SSR slides 19 mm/s, according to the later physical model and experimental results), achieving a same friction reduction as submersion in a water tank. This is shown as the 'local wetting' condition in Fig. 1E."

Fig. 1 ... (E) ... The seven tests have different durations due to different friction values, and therefore we adjusted their time period to the same length for clarity (labelled as normalised time).

Specific comment 2: Both Fig. 1A and Fig. 2A shows the structure and function of the sliding suction of snail. However, from my side, the Fig. 2A may not be necessary because the locomotion of the robot is obvious different from the snail.

Reply: We thank you for this suggestion. We agree that some elements of the snails' mucus secretion mechanism are repeated in Figs 1A and 2A. In Fig. 2A we illustrate the water secretion and spreading method in our design, since this aspect was not included in the former section but important to the sliding of the suction robot. For increased clarity, we have replaced the previous upper diagram in Fig 2A (the snail mucus secretion mechanism) with a rendered image of the water secretion system to clearly show how we achieve the automation of water secretion in this design (page 7, line 179).

Edit: "... We design a water secreting system as shown in Fig. 2A. ..."

Fig. 2. Design of the SSR. (A) The design of water secreting system. ...

Specific comment 3: In Fig. 1E, the scale label of time is missing.

Reply: Thank you for this point. Fig. 1E (as shown in the reply to Specific comment 1) contains the measured sliding friction forces of seven conditions: combinations of hydrophilic/hydrophobic suction cup and dry/water/detergent, and the local wetting condition. During the experiment, we used a linear stage to pull a tensile spring with a constant speed, thereby gradually increasing the pulling force applied to the suction cup. The conditions with high friction (i.e., hydrophobic suction cup in dry and water conditions, and hydrophilic suction cup in dry condition) took longer to begin to slide, while other low friction conditions took a much shorter time to begin to slide. For clarity, we adjusted all the seven conditions to the same flow time duration, to make the sliding friction force easier to understand. We have made it clearer in the main text (page 20, line 558) and have changed the label to “normalised time” in the figure legend (as shown above).

Edit: “... The seven tests have different durations due to different friction values, and therefore we adjusted their time period to the same length for clarity (labelled as normalised time). ...”

Reviewer #2,

General comment: Thank you for your hard work to address my concern. At this stage, I have no more questions.

Reply: We appreciate your efforts reviewing our manuscript. We believe the quality of the manuscript has been significantly improved.

REVIEWERS' COMMENTS

Reviewer #1 (Remarks to the Author):

The authors have well addressed my concerns and comments.